# Impurity Knight shift in quantum dot Josephson junctions

Luka Pavešić[1,2], Marta Pita-Vidal[3], Arno Bargerbos[3] and Rok Žitko[1,2⋆]

**1** Jožef Stefan Institute, Jamova 39, SI-1000 Ljubljana, Slovenia
**2** Faculty of Mathematics and Physics, University of Ljubljana,
Jadranska 19, SI-1000 Ljubljana, Slovenia
**3** QuTech and Kavli Institute of Nanoscience, Delft University of Technology,
2600 GA Delft, The Netherlands

⋆ rok.zitko@ijs.si

## Abstract

Spectroscopy of a Josephson junction device with an embedded quantum dot reveals the presence of a contribution to level splitting in external magnetic field that is proportional to $\cos\phi$, where $\phi$ is the gauge-invariant phase difference across the junction. To elucidate the origin of this unanticipated effect, we systematically study the Zeeman splitting of spinful subgap states in the superconducting Anderson impurity model. The magnitude of the splitting is renormalized by the exchange interaction between the local moment and the continuum of Bogoliubov quasiparticles in a variant of the Knight shift phenomenon. The leading term in the shift is linear in the hybridisation strength $\Gamma$ (quadratic in electron hopping), while the subleading term is quadratic in $\Gamma$ (quartic in electron hopping) and depends on $\phi$ due to spin-polarization-dependent corrections to the Josephson energy of the device. The amplitude of the $\phi$-dependent part is largest for experimentally relevant parameters beyond the perturbative regime where it is investigated using numerical renormalization group calculations. Such magnetic-field-tunable coupling between the quantum dot spin and the Josephson current could find wide use in superconducting spintronics.

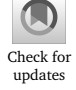

# 1 Introduction

The phenomenon of Knight shift was first observed as a shift of the nuclear magnetic resonance frequencies for atoms in a metal compared with the same atoms in a nonmetallic compound [1]. The shifts were found to be proportional to the amplitude of the hyperfine structure splitting and hence to the strength of the nucleus-electron coupling [1,2]. Knight-shift measurements have since become one of the principal local probe techniques for studying condensed-matter systems [3]. Among their many applications we find the studies of magnetic impurity effects in metals [4] and the testing of the Bardeen-Cooper-Schrieffer (BCS) theory of superconductivity [5–9].

The Knight shift is due to the nucleus-electron hyperfine interaction [2], which is essentially an exchange interaction between the nuclear spin and the electron total angular momentum [10]. By analogy, a similar shift is expected in any system where a local moment is coupled to itinerant particles by exchange coupling. In quantum impurity models, such as the Kondo model and the single-impurity Anderson model (SIAM) for a magnetic impurity in a metallic bath, the Knight shift manifests as a negative shift of the impurity $g$-factor proportional to the Kondo exchange coupling $J$ and the host density of states $\rho$, so that to first order in coupling one has $g_{\text{eff}} = g\left(1 - \frac{1}{2}\rho J\right)$ [11,12],[1] where $g$ is the bare impurity $g$-factor, and $\rho J$ can be expressed in terms of the SIAM parameters as $\rho J = 8\Gamma/\pi U$ with the hybridisation strength $\Gamma$ and the electron-electron repulsion $U$ [13].

---

[1]This expression holds for the case where the Pauli paramagnetism in the host is neglected and the $g$-factor renormalization is a purely dynamic effect due to spin-flip scattering.

In this work we investigate how the *g*-factor is renormalized for an impurity described by the SIAM with a superconducting bath, more specifically for a quantum dot (QD) embedded in a Josephson junction between two superconductors (SCs) with arbitrary gauge-invariant phase difference $\phi$. This is motivated by presented measurements on a QD Josephson junction device based on a semiconductor nanowire embedded in a transmon circuit. These show the presence of a phase-dependent contribution to Zeeman splitting.

We focus on the quantity $\kappa$, the renormalization (impurity Knight shift) factor, defined through $g_{\text{eff}} = g(1-\kappa)$, which is also a measure of the degree of Kondo screening (degree of compensation) [14]. The value of $\kappa$ ranges from 0 for a fully decoupled spin to 1 for a fully compensated spin [14]. Two key results are presented. First, to lowest order in electron hopping, $\kappa$ is found to depend linearly on $\Gamma$. This result is at variance with that found for the Kondo model with a SC bath, where the dependence is quadratic in the exchange coupling $J$ [14]; we will comment on the origin of this difference in the discussion section. Second, and more importantly, $\kappa$ depends on the phase difference between the SCs. This effect is caused by the pair-hopping processes, the very same ones that also produce the Josephson supercurrent [15–19], and to lowest order in electron hopping it is quadratic in $\Gamma$. For experimentally relevant parameters beyond the perturbative regime the magnitude of the phase-dependent term will be quantified using numerical renormalization group (NRG) calculations [20–27]. The physical origin of the $\phi$-dependent contribution to the impurity Knight shift implies the presence of a field-tunable coupling between the Josephson current and the impurity spin. The effect is large and could be used for applications in quantum devices and superconducting spintronics [28–34].

## 2 Experimental evidence

We first present the experimental evidence as motivation. The phase dependence of the Zeeman splitting has been observed in recent experiments where the direct spectroscopy of spin-split Andreev levels has been performed in a QD with SC leads [35]. The device was tuned to a spin-1/2 ground state with an unpaired quasiparticle and excitations were induced by applying a microwave drive to the central gate electrode of the QD. This induces direct transitions between the two branches in the spin doublet subspace described by the following potential energy [35]:

$$U(\phi) = E_0 \cos\phi - E_{\text{SO}}\,\boldsymbol{\sigma} \cdot \mathbf{n} \sin\phi + \frac{1}{2} g\left[1 - \kappa(\phi)\right]\mu_B \boldsymbol{\sigma} \cdot \mathbf{B}, \tag{1}$$

where $\mathbf{n}$ is a unit vector along the polarization direction set by the spin-orbit interaction, and $E_{\text{SO}}$ and $E_0$ are the spin-dependent and spin-independent contributions to the Cooper pair tunneling rate [35]. If a $\cos\phi$ term is present in $\kappa$, so that $\kappa = \bar{\kappa} - \frac{\Delta_\kappa}{2}\cos\phi$, we find for magnetic field applied along $\mathbf{n}$

$$U(\phi) = \left(E_0 \pm \frac{E_Z \Delta_\kappa}{4}\right)\cos\phi \mp E_{\text{SO}} \sin\phi \pm \frac{E_Z(1-\bar{\kappa})}{2}. \tag{2}$$

The transition frequency is given by the difference:

$$E_{\uparrow\downarrow} = E_Z(1-\bar{\kappa}) - 2E_{\text{SO}} \sin\phi + \frac{E_Z \Delta_\kappa}{2}\cos\phi. \tag{3}$$

The experimental results are presented in Fig. 1(a). The plots show the spin-flip frequency as a function of the flux through the SQUID which in turn controls the phase difference across the Josephson junction [35]. The field dependencies of the different contributions are shown in Fig. 1(b). The average spin-flip frequency increases as a linear function of the magnetic

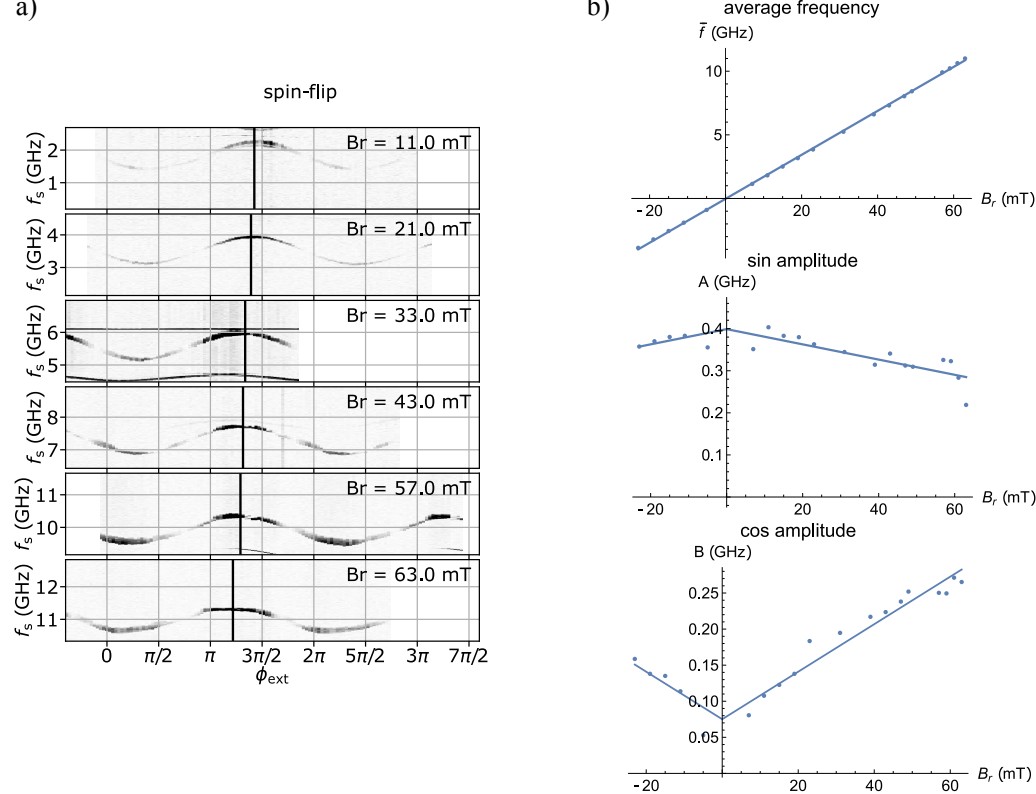

Figure 1: **a**) Spin-flip spectroscopy in a quantum dot with superconducting leads: measured flux dependence of the $|\downarrow\rangle \longleftrightarrow |\uparrow\rangle$ transition for a range of applied magnetic fields. The vertical lines mark the positions of maximal frequency. The field is applied parallel to the spin-orbit coupling direction. **b**) Decomposition of the curves into constant, sine and cosine terms, $f = \bar{f} + A\sin\phi + B\cos\phi$.

field. The $\phi$-dependent terms have amplitudes that are also linear in field. For the sine term, associated with the spin-orbit coupling (SOC), this represents a field-dependent correction that most likely arises from the orbital effects ($E_{SO}$ is generated by cotunneling through high-energy orbitals in the presence of SOC [35]). For the cosine term, the linear dependence is the impurity Knight shift which is the topic of this work. It can be noted that the curve has an offset at zero field. This is due to hysteresis in the flux axis (flux was swept in one direction for the $B > 0$ part, and in the opposite direction for the $B < 0$ part). Nevertheless, the linear field-dependence is clearly demonstrated.

## 3 Model

To model the phenomenon described above we consider a Josephson junction QD [36–46], i.e., a system composed of three parts: a QD and two SCs, see Fig. 2(a). We split the Hamiltonian as $H = H_0 + H_1$, where $H_0$ describes the subsystems in isolation, while $H_1$ describes the coupling

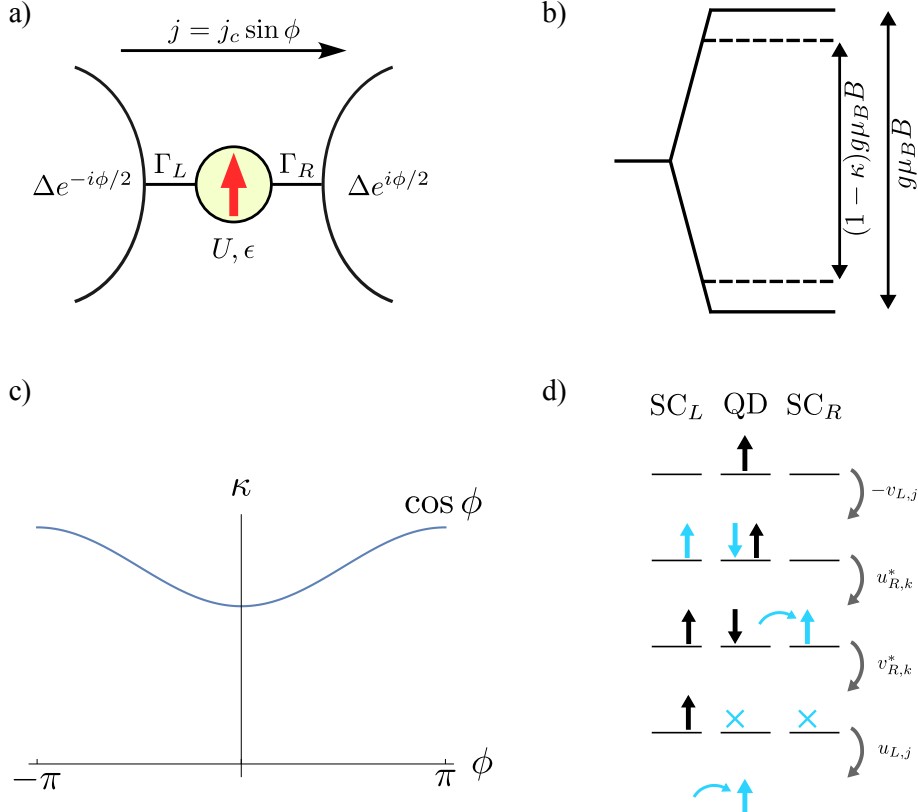

Figure 2: **a**) Schematic representation of the setup: a quantum dot is embedded in the Josephson junction between two superconducting contacts with the gauge-invariant phase difference $\phi$. In the presence of external magnetic field $B$, the quantum dot spin experiences phase-dependent Zeeman splitting, i.e., there is a coupling between the local moment and the supercurrent. **b**) The impurity Knight shift factor $\kappa$ quantifies the reduction of the magnitude of the Zeeman splitting due to the coupling to the contacts $\Gamma$. **c**) For weak and moderate $\Gamma$, $\kappa$ is a harmonic function of $\phi$: $\kappa(\phi) = \bar{\kappa}\left(1 - \frac{\Delta_\kappa}{2}\cos\phi\right)$. **d**) Leading phase-dependent contribution to $\kappa$: fourth-order pair transfer process with the amplitude proportional to $e^{i\phi}$ where an electron pair is transferred from left to right superconducting contact. The blue colored arrows indicate the elements that tunnel in a given step of the process. The amplitude for the process is the product of the four superconducting coherence factors indicated next to the grey arrows.

between them. We write

$$H_0 = H_{\text{imp}} + H_{\text{SC}}^{(L)} + H_{\text{SC}}^{(R)},$$

$$H_{\text{imp}} = \epsilon\hat{n} + U\hat{n}_\uparrow\hat{n}_\downarrow + E_Z\frac{1}{2}\left(\hat{n}_\uparrow - \hat{n}_\downarrow\right)$$

$$= \epsilon_\uparrow\hat{n}_\uparrow + \epsilon_\downarrow\hat{n}_\downarrow + U\hat{n}_\uparrow\hat{n}_\downarrow,$$

$$H_{\text{SC}}^{(\beta)} = \sum_{n,\sigma}\epsilon_{n\sigma}c_{\beta,n\sigma}^\dagger c_{\beta,n\sigma} + \sum_n\left(\Delta e^{i\phi_\beta}c_{\beta,n\uparrow}^\dagger c_{\beta,n\downarrow}^\dagger + \text{H.c.}\right).$$

(4)

The electron annihilation operators are denoted by $d_\sigma$ for the QD and $c_{\beta,n\sigma}$ for the SCs. The index $\beta = L, R$ enumerates the two SCs, while $n\uparrow$ and $n\downarrow$ denote the time-reversal-conjugate pairs of states. In $H_{\text{imp}}$, $\epsilon$ is the impurity energy level, $\hat{n}_\sigma = d_\sigma^\dagger d_\sigma$ are the impurity occupancy operators with $\hat{n} = \hat{n}_\uparrow + \hat{n}_\downarrow$, and $E_Z = g\mu_B B$ is the (bare) impurity Zeeman energy, where $g$

is the atomic Landé $g$-factor, $\mu_B$ is the Bohr magneton, and $B$ is the external magnetic field. In the alternative form, the spin-dependent levels are $\epsilon_\uparrow = \epsilon + g\mu_B B/2$ and $\epsilon_\downarrow = \epsilon - g\mu_B B/2$. We will mainly focus on the case of a half-filled QD with $\epsilon = -U/2$. In $H_{SC}^{(\beta)}$, the energy levels in SCs are $\epsilon_{n,\uparrow} = \epsilon_n + g_S\mu_B B_S/2$ and $\epsilon_{n,\downarrow} = \epsilon_n - g_S\mu_B B_S/2$, where $g_S$ is the atomic Landé $g$-factor of the SC material, and $B_S$ is the field inside the SC. In the following, we set $B_S = 0$, i.e., we assume that the magnetic field does not penetrate in the SCs due to the Meissner effect. This is a good approximation for large SC contacts; in Sec. 7.5 we briefly discuss the field effects in ultrasmall SC islands [47,48] and thin SC layers with in-plane magnetic field [49]. The magnitude of the order parameter $\Delta$ is taken equal in both SCs, while the phases are $\phi_L = \phi/2$ and $\phi_R = -\phi/2$. The normal-state density of states is assumed constant and the band extends from $-D$ to $D$, so that $\rho = \frac{1}{2D}$. The chemical potential is set to $\mu = 0$. In other words, the occupied states extend from $-D$ to $0$, the empty states from $0$ to $D$.

The coupling of the impurity to the SCs is through single-electron hopping:

$$H_1 = H_{1L} + H_{2R}, \quad \text{with} \quad H_{1\beta} = \frac{V_\beta}{\sqrt{N}} \sum_{n\sigma} d_\sigma^\dagger c_{\beta,n\sigma} + \text{H.c.} \tag{5}$$

Here $n$ ranges over all $N$ levels in each SC. The hybridisation strengths can be expressed as constants

$$\Gamma_\beta = \pi\rho V_\beta^2. \tag{6}$$

The spectrum of this model has a number of discrete levels below the continuum of excitations. These discrete (subgap) states are known as Yu-Shiba-Rusinov states (in the $U \gg \Delta$ regime) [50–52] and as proximity-induced states (or Andreev bound states) (in the $U \ll \Delta$ regime) [53], with a sharp transition between the two at $U = 2\Delta$ in the $\Gamma \to 0$ limit [54] and generally a smooth crossover between the two regimes for non-zero $\Gamma$. In this work we focus on the spin-doublet Yu-Shiba-Rusinov states (for large $U$) or odd-parity Andreev states with a single trapped quasiparticle (for small $U$), without being concerned with the question whether these levels are the ground or the excited states of the system; we may, for example, assume that the parity life-time is long enough for the experiment under discussion [46,55].

It is convenient to rewrite the Hamiltonian in terms of Bogoliubov quasiparticle operators $b_{\beta,n\sigma}$. The Bogoliubov quasiparticle states are excitations above the BCS ground state defined through

$$\begin{aligned} c_{\beta,n\uparrow} &= u_{\beta,n}^* b_{\beta,n\uparrow} + v_{\beta,n} b_{\beta,n\downarrow}^\dagger, \\ c_{\beta,n\downarrow}^\dagger &= u_{\beta,n} b_{\beta,n\downarrow}^\dagger - v_{\beta,n}^* b_{\beta,n\uparrow}, \end{aligned} \tag{7}$$

and

$$\begin{aligned} c_{\beta,n\downarrow} &= u_{\beta,n}^* b_{\beta,n\downarrow} - v_{\beta,n} b_{\beta,n\uparrow}^\dagger, \\ c_{\beta,n\uparrow}^\dagger &= u_{\beta,n} b_{\beta,n\uparrow}^\dagger + v_{\beta,n}^* b_{\beta,n\downarrow}. \end{aligned} \tag{8}$$

$u_n$ is the amplitude of the particle-like component of the quasiparticle, $v_n$ that of the hole-like component. We use the following phase convention for these coherence factors:

$$u_{\beta,n} = \sqrt{\frac{1}{2}\left(1 + \frac{\epsilon_n}{\xi_n}\right)}, \quad v_{\beta,n} = e^{i\phi_\beta}\sqrt{\frac{1}{2}\left(1 - \frac{\epsilon_n}{\xi_n}\right)}, \tag{9}$$

where the quasiparticle energies are $\xi_n = \sqrt{\epsilon_n^2 + \Delta^2}$. In this language, the SC Hamiltonians are

$$H_{SC}^{(\beta)} = \sum_{n\sigma} \xi_n b_{\beta,n\sigma}^\dagger b_{\beta,n\sigma}. \tag{10}$$

The index $n$ runs over all $N$ levels in each bath, for $\epsilon_n$ of either sign. We have dropped the constant terms since only the excitation energies are important for what follows.

The tunneling parts can be rewritten as

$$H_{1\beta}/(V/\sqrt{N}) = \sum_n \Big[ d_\uparrow^\dagger \big( u_{\beta,n}^* b_{\beta,n\uparrow} + v_{\beta,n} b_{\beta,n\downarrow}^\dagger \big) + d_\downarrow^\dagger \big( u_{\beta,n}^* b_{\beta,n\downarrow} - v_{\beta,n} b_{\beta,n\uparrow}^\dagger \big)$$
$$+ \big( u_{\beta,n} b_{\beta,n\uparrow}^\dagger + v_{\beta,n}^* b_{\beta,n\downarrow} \big) d_\uparrow + \big( u_{\beta,n} b_{\beta,n\downarrow}^\dagger - v_{\beta,n}^* b_{\beta,n\uparrow} \big) d_\downarrow \Big]. \tag{11}$$

The hopping processes do not conserve the number of particles, only its parity. An electron hopping from the impurity level to the bath can either become a quasiparticle, or annihilate one. The latter process can be understood as the recombination of an existing Bogoliubov quasiparticle and the hopping electron of the opposite spin into a Cooper pair.

## 4 Problem formulation

In the presence of magnetic field the lowest-lying doublet splits by $E_\uparrow - E_\downarrow$ and we define the effective impurity $g$-factor as

$$g_{\text{eff}} = \frac{E_\uparrow - E_\downarrow}{\mu_B B}. \tag{12}$$

Here $E_\sigma$ designate the energies of the eigenstates of the total spin operator with $S_z = \sigma$. For a free impurity, $g_{\text{eff}} = g$. The renormalization factor $\kappa$ is a measure of the negative deviation of $g_{\text{eff}}$ from $g$:

$$\kappa = \frac{g - g_{\text{eff}}}{g}. \tag{13}$$

We thus need to compute the correction $\Delta E_\sigma$ to the eigenenergies $E_\sigma$ due to the impurity-bath coupling $H_1$ and take their difference scaled by $E_Z = g\mu_B B$:

$$\kappa = \frac{\Delta E_\downarrow - \Delta E_\uparrow}{E_Z}. \tag{14}$$

The goal of this work is to calculate this quantity perturbatively and to verify these results (and extend them to higher values of $\Gamma$) with a numerical solution using the NRG.

In SIAM, there are two microscopic mechanisms which renormalize the Zeeman splitting, spin and charge fluctuations (see also the discussion in Sec. 7.3). The spin fluctuations increase the admixture of a wavefunction component where the impurity forms a singlet with a quasiparticle, with equal contributions of $S_{z,\text{imp}} = +1/2$ and $S_{z,\text{imp}} = -1/2$. The charge-fluctuation mechanism means that in the spin-doublet state there are components of the wavefunction with configurations where the impurity is unoccupied or doubly occupied (e.g. as virtual states during spin-flip events), hence $S_{z,\text{imp}} = 0$. Both mechanisms reduce $g$, their relative importance depends on the value of the $U/\Delta$ ratio.

The quantity $\kappa$ is also proportional to the scalar product between the impurity and bath spin, i.e., it represents the degree of Kondo screening by the particles in the bath [14]. It can furthermore be related to the expectation value of the impurity spin operator $\hat{S}_{z,\text{imp}}$ [14]:

$$\kappa = 1 - 2\langle \psi | \hat{S}_{z,\text{imp}} | \psi \rangle = 1 - 2\,\text{Tr}\big[ \hat{\rho}_{\text{imp}} \hat{S}_{z,\text{imp}} \big], \tag{15}$$

where $\hat{\rho}_{\text{imp}} = \text{Tr}_{\text{bath}}[\hat{\rho}]$ is the impurity density matrix obtained by tracing out the SC bath degrees of freedom. The definitions in Eq. (14) and (15) are fully equivalent and both may be used in numerical calculations using the NRG; see also Sec. 6 for technical details. The operator $\hat{S}_{z,\text{imp}}$ is diagonal in the impurity basis $i \in \{\uparrow, \downarrow, 0, 2\}$, thus

$$\kappa = 1 - P_\uparrow + P_\downarrow, \tag{16}$$

where $P_i$ are the expectation values of the projection operators $\hat{P}_i = |i\rangle\langle i|$ in the total spin-up doublet ground state of the problem. Note that there is a sum rule $\sum_i P_i = 1$. The spin fluctuations are accounted for through non-zero $P_\downarrow$ (at the expense of $P_\uparrow$), and the charge fluctuations through the reduction of $P_\uparrow + P_\downarrow$ due to non-zero $P_0$ and $P_2$, following the sum rule. In Sec. 6 we will use this observation to disentangle the various contributions to the impurity Knight shift.

## 5 Perturbative calculation

In the following, we treat the hopping $H_1$ as a perturbation to $H_0$. We use the Rayleigh-Schrödinger perturbation theory (PT) [56] to compute the corrections to second and fourth order in $H_1$ (odd-order contributions are all zero) [37] using the projector operator approach with a symbolic algebra system [57, 58]. The $n$-th order correction to $\kappa$ is defined as

$$\kappa^{(n)} = \frac{\Delta E_\downarrow^{(n)} - \Delta E_\uparrow^{(n)}}{E_Z} \,. \tag{17}$$

### 5.1 Zeroth order

The ground state of $H_0$ is the Zeeman-split pair

$$
\begin{aligned}
|\psi_\uparrow\rangle &= |\uparrow\rangle \otimes |\text{BCS}\rangle = d_\uparrow^\dagger |0\rangle \otimes |\text{BCS}\rangle \,, \\
|\psi_\downarrow\rangle &= |\downarrow\rangle \otimes |\text{BCS}\rangle = d_\downarrow^\dagger |0\rangle \otimes |\text{BCS}\rangle \,,
\end{aligned}
\tag{18}
$$

with energies $E_\uparrow = \epsilon + E_Z/2 = \epsilon_\uparrow$ and $E_\downarrow = \epsilon - E_Z/2 = \epsilon_\downarrow$. Here $|0\rangle$ is the empty state of the QD, $|\sigma\rangle = d_\sigma^\dagger |0\rangle$ are the singly occupied states; $|\text{BCS}\rangle$ is the ground state of both superconductors in isolation (i.e., a product state of two BCS states, one for each superconductor) which is annihilated by all Bogoliubov quasiparticle operators: $b_{\beta,n\sigma}|\text{BCS}\rangle \equiv 0$.

### 5.2 Second order

In the second-order PT, the energy shifts are

$$\Delta E_\sigma^{(2)} = -\sum_k \frac{|V_{k\sigma}|^2}{\mathcal{E}_{k\sigma}} \,, \tag{19}$$

with $k$ running over all intermediate states. The electron hopping matrix elements are defined as $V_{ij} = \langle i|H_1|j\rangle$, while $\mathcal{E}_{k\sigma} = E_k - E_\sigma$ is the energy of the intermediate state $k$ with respect to the energy of the initial state $\psi_\sigma$.

Only two types of processes are possible in the second-order PT. Either a Cooper pair splits and one electron tunnels into the impurity level, or the impurity electron tunnels into the bath where it materializes as a Bogoliubov quasiparticle. The intermediate states $|\psi_{\beta,n\sigma}^{A/B}\rangle$, their energies, and tunneling matrix elements are:

$$|\psi_{\beta,n\sigma}^A\rangle = |2\rangle \otimes b_{\beta,n\sigma}^\dagger |\text{BCS}\rangle \,, \quad E_{A,n\sigma} = \epsilon_{\bar{\sigma}} + U + \xi_n \,, \quad \langle\psi_{\beta,n\sigma}^A|h_{1\beta}|\psi_\sigma\rangle = v_{\beta,n} \,, \tag{20}$$

$$|\psi_{\beta,n\sigma}^B\rangle = |0\rangle \otimes b_{\beta,n\sigma}^\dagger |\text{BCS}\rangle \,, \quad E_{B,n\sigma} = -\epsilon_\sigma + \xi_n \,, \quad \langle\psi_{\beta,n\sigma}^B|h_{1\beta}|\psi_\sigma\rangle = u_{\beta,n} \,, \tag{21}$$

where $h_{1\beta} = H_{1\beta}/(V_\beta/\sqrt{N})$ is the normalized tunneling Hamiltonian of Eq. (11). The sign convention for the doubly occupied state is $|2\rangle = d_\downarrow^\dagger d_\uparrow^\dagger |0\rangle$. $\bar{\sigma}$ denotes the spin opposite to $\sigma$.

The energy shift is found to be

$$\Delta E_\sigma^{(2)} = -\sum_{\beta,n} \frac{V_\beta^2}{N}\left[\frac{|v_n|^2}{E_{A,n\sigma}} + \frac{|u_n|^2}{E_{B,n\sigma}}\right]. \tag{22}$$

Each SC contributes independently and additively. We also note that in the second-order PT the phase factors (contained in $v_n$) play no role, because the result only contains the absolute values of matrix elements.

At the impurity particle-hole symmetric point, $\epsilon = -U/2$, we have $E_{A,n\sigma} = E_{B,n\sigma} \equiv E_{n\sigma}$ with $E_{n\uparrow} = U/2 - E_Z/2 + \xi_n$ and $E_{n\downarrow} = U/2 + E_Z/2 + \xi_n$. Thus

$$\begin{aligned}
\kappa^{(2)} &= \frac{\Delta E_\downarrow^{(2)} - \Delta E_\uparrow^{(2)}}{E_Z} = -\sum_\beta \frac{V_\beta^2}{E_Z N}\sum_n\left(|v_n|^2 + |u_n|^2\right)\left(\frac{1}{E_{n\downarrow}} - \frac{1}{E_{n\uparrow}}\right) \\
&= \sum_\beta \frac{V_\beta^2}{E_Z N}\sum_n\left(\frac{1}{E_{n\uparrow}} - \frac{1}{E_{n\downarrow}}\right).
\end{aligned} \tag{23}$$

The sum runs over all quasiparticle levels $n$ and can be converted to an integration over the kinetic energies $\epsilon$. The prescription is the same as for a metal, because the distribution of levels is assumed constant:

$$\frac{1}{N}\sum_n \rightarrow \rho\int d\epsilon. \tag{24}$$

The difference $E_{n\uparrow}^{-1} - E_{n\downarrow}^{-1}$ can be expressed as

$$E_{n\uparrow}^{-1} - E_{n\downarrow}^{-1} = \frac{E_Z}{-E_Z^2/4 + (U/2 + \sqrt{\Delta^2 + \epsilon})^2}. \tag{25}$$

The $E_Z^2$ contribution in the denominator is always negligible and may be dropped. Thus

$$\begin{aligned}
\kappa^{(2)} &= \sum_\beta \rho V_\beta^2 I^{(2)}(U, \Delta, D) = \frac{1}{\pi}\left(\sum_\beta \Gamma_\beta\right)I^{(2)}(U, \Delta, D), \\
&\text{with}\quad I^{(2)}(U, \Delta, D) = \int_{-D}^{D}\frac{d\epsilon}{(U/2 + \sqrt{\Delta^2 + \epsilon^2})^2}.
\end{aligned} \tag{26}$$

The renormalization factor $\kappa^{(2)}$ is proportional to the total hybridisation strength,

$$\Gamma = \Gamma_L + \Gamma_R. \tag{27}$$

The integral may be transformed into dimensionless form by factoring out $1/\Delta$:

$$I^{(2)}(U, \Delta, D) = \frac{1}{\Delta}i^{(2)}(u, d), \tag{28}$$

where $u = U/\Delta$ and $d = D/\Delta$ are the dimensionless interaction strength and the dimensionless bandwidth, and

$$i^{(2)}(u, d) = \int_{-d}^{d}\frac{dx}{\left[u/2 + \sqrt{1 + x^2}\right]^2}. \tag{29}$$

The integral can be evaluated in closed form:

$$i^{(2)}(u, d) = \frac{8}{w^2}\left\{4\sqrt{-w}\left[\arctan\left(\frac{u+2}{\sqrt{-w}}\right) - \arctan\left(\frac{u + 2\sqrt{1 + d^2} - 2d}{\sqrt{-w}}\right)\right] \right. \\
\left. + \frac{duw(2\sqrt{1 + d^2} - u)}{4d^2 - w}\right\}, \tag{30}$$

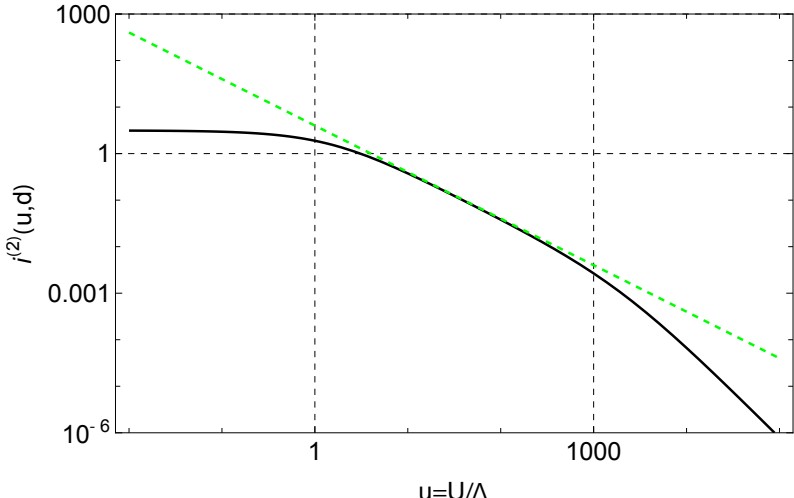

Figure 3: $U$-dependence of the second-order contribution on log-log scale. The full black line is $i^{(2)}(u,d)$ from Eq. (30) as a function of $u = U/\Delta$ at fixed bandwidth $d = D/\Delta = 10^3$. The green dashed line corresponds to the $4/u$ asymptotic form in the $\Delta < U < D$ regime.

where $w = u^2 - 4$. In the infinite-bandwidth limit, this simplifies to

$$i^{(2)}(u, d \to \infty) = \frac{8}{w^2} \left\{ 4\sqrt{-w} \left[ \arctan\left( \frac{u+2}{\sqrt{-w}} \right) - \arctan\left( \frac{u}{\sqrt{-w}} \right) \right] + \frac{uw}{2} \right\}. \tag{31}$$

The standard branch cuts apply here. In particular, for $u > 2$ the following form may be used instead:

$$i^{(2)}(u, d \to \infty) = \frac{4}{w^{3/2}} \left[ u\sqrt{w} + 8\,\text{atanh}\sqrt{\frac{u+2}{u-2}} - 8\,\text{atanh}\frac{u}{\sqrt{w}} \right]. \tag{32}$$

For small $u$ we find $i^{(2)}(u, d \to \infty) \approx \pi - 2u$, while for large $u$ we find $i^{(2)}(u, d \to \infty) \approx 4/u$; if $d$ is finite, this $4/u$ scaling holds up to $u \sim d$. Furthermore, $u(2, d \to \infty) = 4/3$. For general $u$ and finite $d$, the function $i^{(2)}(u, d)$ is plotted in Fig. 3.

From these expressions, we infer the following asymptotic results. For low $U$ in the infinite bandwidth limit,

$$I^{(2)}(U, \Delta) \approx \frac{1}{\Delta} \left( \pi - \frac{2U}{\Delta} \right). \tag{33}$$

For $U \ll \Delta$, this gives

$$\kappa^{(2)} = \sum_\beta \rho V_\beta^2 \frac{\pi}{\Delta} = \sum_\beta \frac{\Gamma_\beta}{\Delta} = \sum_\beta \frac{\pi}{8} \rho J_\beta \frac{U}{\Delta}, \tag{34}$$

where the Kondo coupling constant $J_\beta$ for the SC $\beta$ is defined through $\rho J_\beta = 8\Gamma_\beta/\pi U$.

For $\Delta \ll U \ll D$, we find

$$I^{(2)}(U, 0) = \int_{-D}^{D} \frac{1}{(U/2 + \epsilon)^2} d\epsilon = \frac{4D}{U(U/2 + D)} \approx \frac{4}{U}. \tag{35}$$

In this regime we recover the result for the normal-state case:

$$\kappa^{(2)} = \sum_\beta V_\beta^2 \rho \frac{4}{U} = \sum_\beta \frac{1}{2} \frac{8}{\pi} \frac{\Gamma_\beta}{U} = \sum_\beta \frac{1}{2} \rho J_\beta. \tag{36}$$

Finally, if $U$ exceeds all other energy scales in the problem, $U \gg \Delta, D$, we find

$$I^{(2)}(U, 0) = \int_{-D}^{D} \frac{1}{(U/2)^2} d\epsilon = \frac{8D}{U^2}, \tag{37}$$

so that

$$\kappa^{(2)} = \sum_{\beta} \frac{8}{\pi} \frac{D\Gamma_{\beta}}{U^2} = \sum_{\beta} \rho J_{\beta} \frac{D}{U}. \tag{38}$$

For $\Delta \ll D$, which is always the case in real systems, there are thus three well-separated regimes depending on the value of the electron-electron repulsion $U$: 1) the weakly-interacting regime for $U \ll \Delta$ with $\kappa \propto \Gamma/\Delta$, 2) the cross-over regime for $\Delta \ll U \ll D$ with $\kappa \propto \Gamma/U$, 3) narrow-band limit for $D \ll U$ with $\kappa \propto \Gamma/U^2$. The first regime is typical of weakly interacting junctions [59–66], the second one of strongly interacting ones [35,46,67]; the third is mostly of academic interest in relation with the Schrieffer-Wolff mapping between the SIAM and the Kondo models [13,21], but would be relevant for flat-band superconductors.

### 5.3 Fourth order

The fourth-order correction has two contributions. The first is

$$\Delta E_{\sigma}^{(4a)} = - \sum_{j,i,k;i\neq\sigma} \frac{V_{\sigma j} V_{ji} V_{ik} V_{k\sigma}}{\mathcal{E}_{j\sigma}\mathcal{E}_{i\sigma}\mathcal{E}_{k\sigma}}, \tag{39}$$

where $i, j, k$ denote the intermediate states. This is a sum of contributions of all hopping processes which start and end in the initial state $\psi_{\sigma}$ after four electron hopping events, without passing through the initial state $\psi_{\sigma}$ (this restriction is only relevant for the sum over $i$). The second contribution is

$$\Delta E_{\sigma}^{(4b)} = \sum_{k} \frac{|V_{k\sigma}|^2}{\mathcal{E}_{k\sigma}} \times \sum_{k} \frac{|V_{k\sigma}|^2}{\mathcal{E}_{k\sigma}^2}. \tag{40}$$

The expression (39) simplifies to a double sum which can be transformed into a double integral by the rule given in Eq. (24). Likewise, Eq. (40) is a product of two energy integrals. We define

$$I^{(4a)}(U, \Delta, D) = \Delta E_{\downarrow}^{(4a)} - \Delta E_{\uparrow}^{(4a)}, \quad I^{(4b)}(U, \Delta, D) = \Delta E_{\downarrow}^{(4b)} - \Delta E_{\uparrow}^{(4b)}, \tag{41}$$

and finally

$$\kappa^{(4)} = \sum_{\beta\beta'} V_{\beta}^2 V_{\beta'}^2 \rho^2 \left[ I_{\beta\beta'}^{(4a)}(U, \Delta, D) + I_{\beta\beta'}^{(4b)}(U, \Delta, D) \right]. \tag{42}$$

In this expression we separated the contributions depending on which leads the electron hops to/from in the process; $\beta$ and $\beta'$ range over $L$ and $R$, such that $LL$ and $RR$ contributions involve two excursions into the same lead, while the more interesting $LR$ and $RL$ involve both leads. Setting $\xi_i = \sqrt{\Delta^2 + \epsilon_i^2}$, the full expressions for the integrals are:

$$\begin{aligned}
I_{LL}^{(4a)} = I_{RR}^{(4a)} = -\int_{-D}^{D} d\epsilon_1 \int_{-D}^{D} d\epsilon_2 \, 8 \big( U \left( 11\xi_1^2 + 20\xi_1\xi_2 + 5\xi_2^2 \right) \\
+ 2(\xi_1 + \xi_2)\left( 6U^2 + 4\xi_1^2 + 3\xi_1\xi_2 + \xi_2^2 \right) + U^3 \big) \times \\
\times \frac{\epsilon_1\epsilon_2 - \xi_1\xi_2 + \Delta^2}{\xi_1\xi_2(\xi_1 + \xi_2)^2(2\xi_1 + U)^2(2\xi_2 + U)^3},
\end{aligned} \tag{43}$$

$$I_{LR}^{(4a)} + I_{RL}^{(4a)} = -\int_{-D}^{D} d\epsilon_1 \int_{-D}^{D} d\epsilon_2 16 \left( U \left( 11\xi_1^2 + 20\xi_1\xi_2 + 5\xi_2^2 \right) \right.$$
$$\left. + 2(\xi_1 + \xi_2)\left( 6U^2 + 4\xi_1^2 + 3\xi_1\xi_2 + \xi_2^2 \right) + U^3 \right) \times$$
$$\times \frac{\epsilon_1\epsilon_2 - \xi_1\xi_2 + \Delta^2 \cos(\phi)}{\xi_1\xi_2(\xi_1 + \xi_2)^2(2\xi_1 + U)^2(2\xi_2 + U)^3}, \tag{44}$$

and

$$I_{\beta\beta'}^{(4b)} = -\int_{-D}^{D} d\epsilon_1 \int_{-D}^{D} d\epsilon_2 \frac{16(4\xi_1 + 2\xi_2 + 3U)}{(2\xi_1 + U)^2(2\xi_2 + U)^3}. \tag{45}$$

The most important new feature here is the $\cos(\phi)$ term in Eq. (44). Its origin are processes with an amplitude that contains a factor such as $v_{L,i}v_{R,j}^* = e^{i\phi}|v_{L,i}| \cdot |v_{R,j}|$. This is only possible when a pair of electrons is transferred across the junction, see Fig. 1(d) for an illustration. The conjugate process for the transfer of a pair in the opposite direction contributes a $e^{-i\phi}$ term, so that the sum of both terms then produces the $\cos\phi$ terms in the final expressions. While $I^{(4a)}$ encompasses true fourth-order processes, $I^{(4b)}$ is obtained as a product of two second-order PT terms, thus only $I^{(4a)}$ depends on the phase difference $\phi$.

We introduce $x_1 = \epsilon_1/\Delta$, $x_2 = \epsilon_2/\Delta$ and factor out $1/\Delta^2$ in front of the integrals, so that

$$I^{(4)}(U, \Delta, D) = \frac{1}{\Delta^2} i^{(4)}(u, d), \tag{46}$$

where $i^{(4)}$ are dimensionless functions of $u = U/\Delta$ and $d = D/\Delta$. In the following we focus on the wide band limit, $d \to \infty$. We single out the $\phi$-dependent part of $i^{(4a)}$, which we denote $i^{(4,\phi)}$. We were unable to find a closed form expression for this function. Noting that for $u \ll 1$ the function $i^{(4,\phi)}$ becomes constant, and that for $u \gg 1$ it decreases as $1/u^2$, we write it as

$$i^{(4,\phi)}(u, \infty) = -\frac{c}{1 + cu^2/32} A(u) \cos(\phi). \tag{47}$$

Here

$$c = 3\pi^2 - 4 - \frac{2}{\sqrt{\pi}} G_{3,3}^{3,2}\left(1 \left| \begin{matrix} -1, 1/2, 1 \\ 0, 0, 0 \end{matrix} \right.\right) \approx 19.7392 \tag{48}$$

is the $u = 0$ asymptotic value of the double integral, where $G$ is the Meijer's $G$-function. $A(u)$ is a function with values of order 1 (a "form function"), such that $A(0) = A(\infty) = 1$, that we plot in Fig. 4(a).

We conclude that the $\phi$-dependent part of the renormalization $\kappa$ takes the following form in the wide-bandwidth limit:

$$\kappa^{(4,\phi)} = -\frac{\Gamma_L\Gamma_R}{\pi^2\Delta^2} \frac{cA(U/\Delta)}{1 + \frac{c}{32}\left(\frac{U}{\Delta}\right)^2} \cos(\phi). \tag{49}$$

The fourth-order contributions to $\kappa$ that do not depend on the phase $\phi$ are small compared with the dominant second-order contribution, thus we discuss them only briefly. We focus on the symmetric case with $V_L = V_R$. We define $i^{(4,x)}$ to be the sum up of all contributions from $i^{(4a)}$ and $i^{(4b)}$ that do not depend on $\phi$. We symmetrize this expression with respect to $x_1$ to obtain a well-behaved convergent integrand. We denote it $i^{(4,x)}$. For $u \ll 1$ it becomes constant, it changes sign at $u = u_0 \approx 2.51$, and for $u \gg 1$ it decreases as a product of $1/u^2$ and some approximately logarithmic factor. We hence write $i^{(4,x)}$ as

$$i^{(4,x)}(u, \infty) = -3c/(1 + u^2)B(u). \tag{50}$$

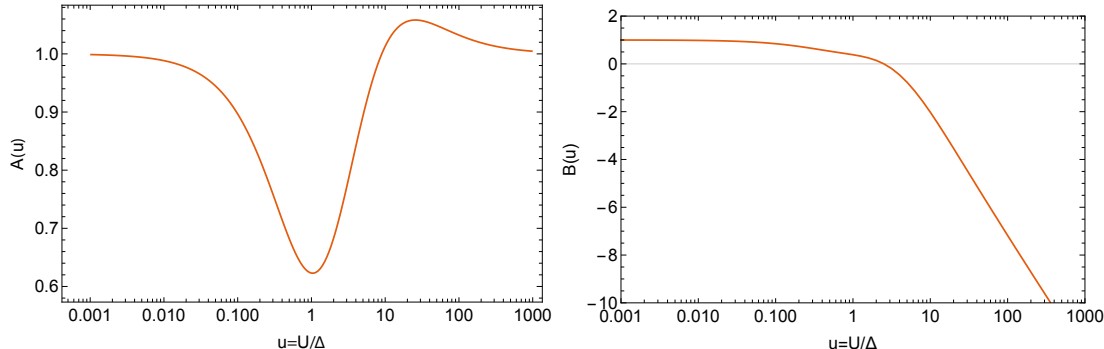

Figure 4: Form functions $A(u)$ and $B(u)$ that determine the detailed dependence of the fourth-order contribution to $\kappa$ on the scaled interaction strength, $u = U/\Delta$, in the infinite bandwidth limit. Left: Function $A(u)$ for $i^{(4,\phi)}$ in Eq. (47). Right: Function $B(u)$ for $i^{(4,x)}$ in Eq. (50).

Here $B(u)$ is a function such that $B(0) = 1$, $B(u_0) = 0$, and with approximately logarithmic asymptotic behaviour for large $u$, as shown in Fig. 4(b). Thus, the fourth order renormalization $\kappa$ that does not depend on $\phi$ takes the following form:

$$\kappa^{(4,x)} = -\frac{(\Gamma/2)^2}{\pi^2 \Delta^2} \frac{3c}{1 + (U/\Delta)^2} B(U/\Delta), \tag{51}$$

for the symmetric case of $\Gamma_L = \Gamma_R = \Gamma/2$.

All analytical calculations are made available in the form of a Mathematica notebook[2] in an online repository [68]. The notebook contains full expressions for the integrands and it allows to reproduce all calculations presented in this work, as well as to calculate $i^{(4,\phi)}$ and $i^{(4,x)}$ for arbitrary parameters.

## 6 Beyond the perturbative regime

Realistic devices are typically operated in the parameter regime where the results of the PT are not adequate, i.e., $\Gamma$ is usually not much smaller than all other scales in the problem ($U$, $\Delta$). The impurity problem can, however, be solved with high precision for arbitrary parameters using an impurity solver such as the numerical renormalization group (NRG). The NRG is based on discretizing the continuum on a logarithmic grid, transforming the Hamiltonian to a tight-binding-chain representation, and iteratively diagonalizing the chain by adding one site (per bath) at each step [20,21]. The results are typically within a few percent of the exact ones. The findings presented here were obtained for the discretization parameter $\Lambda = 2$ (at $\phi = 0$) and $\Lambda = 4$ or $\Lambda = 8$ (with averaging over two shifted discretization grids – $z$-averaging) for general $\phi$, keeping up to 10000 states (or up to a cutoff of 10 units of characteristic energy). The renormalization factor is extracted by performing the calculation at finite $E_Z$, then taking the energy difference between the lowest lying doublet states. The value of $E_Z$ should be taken low enough to be well within the linear Zeeman splitting regime (a fraction of $\Delta$ such as $10^{-2}\Delta$ is a perfectly good choice). The other approach that we employ is to read off $\kappa$ from the matrix elements of $\hat{S}_Z$ at the end of the iteration performed for $E_Z = 0$ according to Eq. (15). The two approaches produce almost perfectly overlapping results. This is because the spin operator $\hat{S}_Z = (1/2)(\hat{n}_\uparrow - \hat{n}_\downarrow)$ is marginal [35].

---

[2]Filename is knight_shift_perturbation_calculation.nb.

a)
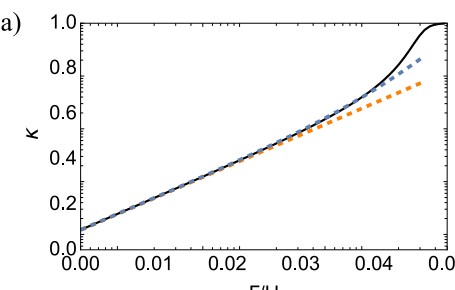

b)
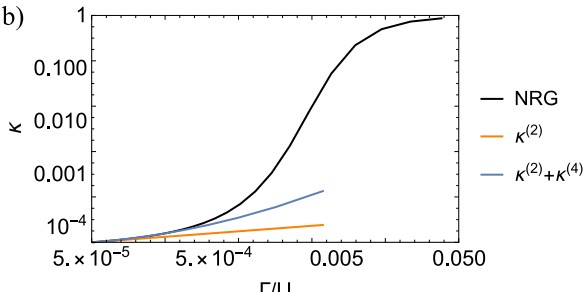

Figure 5: Zeeman renormalisation factor $\kappa$ as a function of the hybridisation strength $\Gamma$ computed using the NRG and compared with the perturbation theory results including up to second or fourth order contributions on **a)** logarithnimc and **b)** linear scales. Here $U/\Delta = 10^4$, $D = 10^5\Delta$, and $\phi = 0$.

## 6.1 $\Gamma$-dependence: from perturbative regime to full spin compensation

We check the domain of validity of second and fourth-order PT results by comparing them to the reference NRG solution. The $\kappa \propto \Gamma$ scaling at very small $\Gamma$ is demonstrated in Fig. 5. The leading $\Gamma^2$ correction captures the deviation from linearity at larger $\Gamma$, but we see that more generally the low-order PT results significantly underestimate $\kappa$ and high-order terms become relevant. For very large values of $\Gamma$, both doublet states approach the edge of the continuum and the energy difference $E_\uparrow - E_\downarrow$ tends toward zero, therefore $g_{\text{eff}} \to 0$ and $\kappa \to 1$. This large-$\Gamma$ asymptotic behaviour holds generally.

For a more systematic overview of the dependence of $\kappa$ on model parameters, in Fig. 6 we plot $\kappa$ as a function of $\Gamma$ for a wide range of $U/\Delta$ ratios. Both panels present the same results, but plotted as a function of $\Gamma/U$ and $\Gamma/\Delta$, respectively. The small-$\Gamma$ asymptotics are clearly visible. In Fig. 6(a) the curves overlap for $U \gg \Delta$ where $\kappa \propto \Gamma/U$. In Fig. 6(b), the curves overlap for $U \ll \Delta$ where $\kappa \propto \Gamma/\Delta$. In general, the line shape depends on the $U/\Delta$ ratio, and there is no universality as in the case of the Kondo model, where $\kappa(J)$ is a universal function of $T_K/\Delta$ with $T_K(J)$ the Kondo temperature [14]; for SIAM, such universality at best holds only in a moderate range of parameters and, in particular, it is not expected for most experimentally relevant parameter sets where typically $U \sim \Delta$ for devices operated in or close to the YSR regime. This also implies that quantitatively reliable results can only be obtained by advanced numerics such as NRG; for convenience, the numerical results presented in the figures of this manuscript are available in tabulated form in a public repository [68].

To estimate the magnitude of the impurity Knight shift in real systems, we recall that $\Gamma/U$ in typical devices is of order 0.1 and note that $\kappa(\Gamma/U = 0.1) \sim 0.05$ for $U/\Delta = 1$ and $\kappa(\Gamma/U = 0.1) \sim 0.15$ for $U/\Delta \sim 10$. It is thus expected that typical values of $\kappa$ are of order 0.1, i.e., the effect is appreciable for realistic model parameters of typical experimental devices.

## 6.2 $U$-dependence: the three interaction strength regimes

The different parameter regimes with respect to $U$ can be better discerned if the results are plotted for a set of fixed $\Gamma/U$ ratios as a function of $U$, see Fig. 7. For the lowest values of $\Gamma/U = 10^{-4}, 10^{-3}$, the system remains in the deep perturbative regime, and the curves basically follow the $U$ dependence established using the second-order perturbation theory in Sec. 5.2: we find $1/U$ scaling for $U > D$, a plateau for $\Delta < U < D$, and $U$ scaling for $U < \Delta$. Non-perturbative effects become manifest for $\Gamma/U \gtrsim 0.01$. The $1/U$ range is replaced by a milder decrease followed by saturation, while at still higher $\Gamma$ the $\kappa(U)$ curves become monotonically increasing. The saturation at large $U$ is expected, since for fixed $\Gamma/U$, $U \gg D$ implies that $D$ rather than $U$ controls the effective bandwidth for the emergence of the local

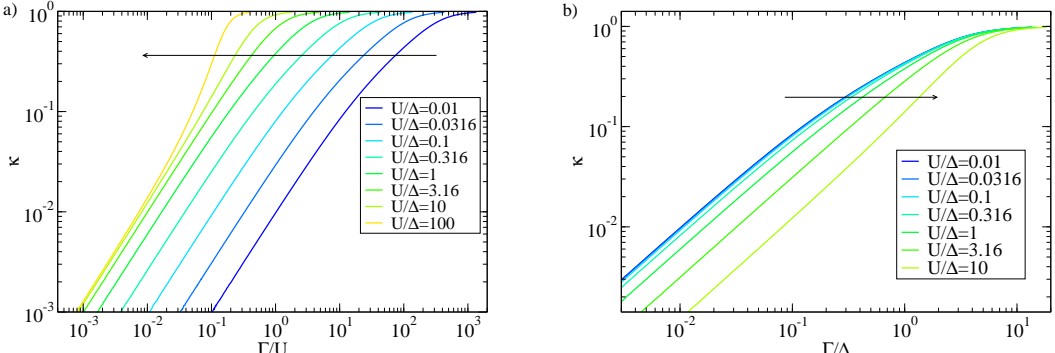

Figure 6: Zeeman renormalisation factor $\kappa$ as a function of $\Gamma$ at fixed $U$, for a range of $U$ spanning from the $U \ll \Delta$ to the $U \gg \Delta$ regime. The horizontal axis is scaled as $\Gamma/U$ (**a**) and $\Gamma/\Delta$ (**b**). The arrows indicate the direction of increasing parameter $U$. Here $D = 10^5\Delta$, $\phi = 0$.

moment [21], while the Kondo exchange $\rho J = 8\Gamma/\pi U$ is constant [13], hence the factor $\kappa$ does not depend on $U$ for $U \to \infty$. The saturation value itself is an increasing function of $\Gamma/U$. The largest value of $\Gamma/U$ presented, 0.1 (dark blue line in Fig. 7), corresponds to an experimentally relevant value, thus that curve can serve to estimate $\kappa$ based on the $U/\Delta$ ratio.

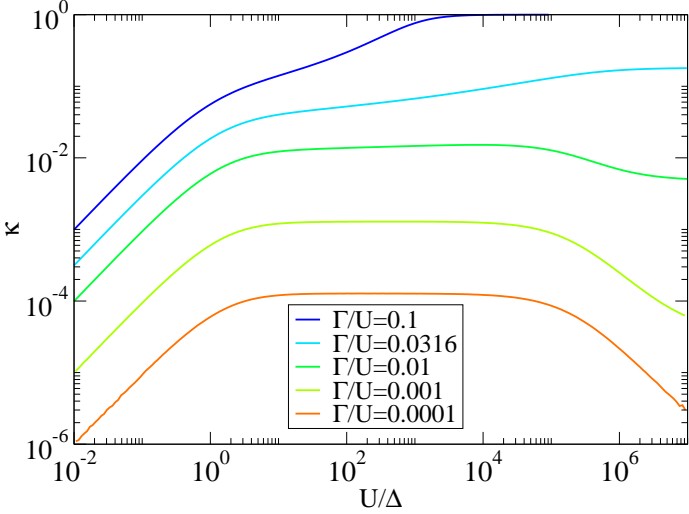

Figure 7: Zeeman renormalisation factor $\kappa$ as a function of $U/\Delta$ for a range of fixed $\Gamma/U$ ratios. Parameters are $D = 10^5\Delta$ and $\phi = 0$.

## 6.3 $\epsilon$-dependence: departure from the particle-hole symmetric point

In Fig. 8 we present the dependence of $\kappa$ on the impurity level position $\epsilon$. The renormalization is the lowest at the particle-hole symmetric point where the exchange interaction is the smallest

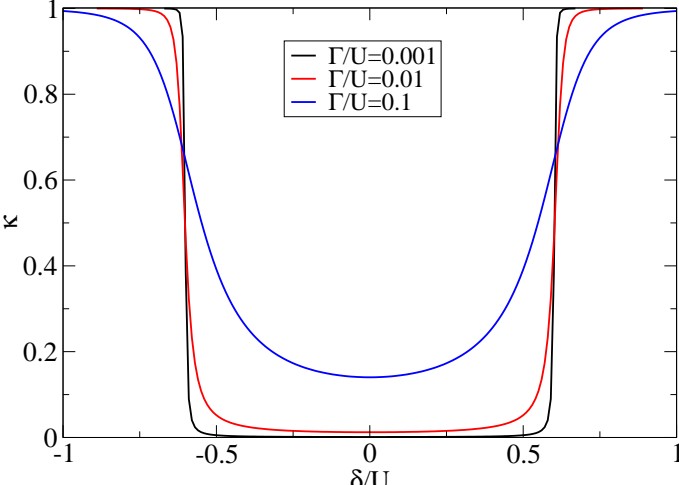

Figure 8: Zeeman renormalisation factor $\kappa$ as a function of $\delta/U$. Here $\delta = \epsilon + U/2$ measures the deviation from the particle-hole symmetric point of the model. Parameters are $U/\Delta = 10$, $D = 10^5\Delta$ and $\phi = 0$.

[13], since

$$\rho J = \frac{2\Gamma}{\pi}\left(\frac{1}{\epsilon + U} - \frac{1}{\epsilon}\right) = \frac{2\Gamma}{\pi}\left(\frac{1}{U/2 + \delta} + \frac{1}{U/2 - \delta}\right) = \frac{2\Gamma}{\pi}\frac{4U}{U^2 - 4\delta^2}. \tag{52}$$

Here $\delta = \epsilon + U/2$ quantifies the deviation from the particle-hole symmetric point (half-filling) at $\delta = 0$. As the charge fluctuations increase for $\delta/U \to \pm 1/2$, the local moment is reduced and $\kappa$ rapidly increases toward 1 (at the same time, the doublet subgap state is pushed towards the continuum of free Bogoliubov states).

## 6.4  $\phi$-dependence

We now turn to the $\phi$-dependence of the factor $\kappa$. Except for very strong hybridisation we expect the dependence to follow a $\cos\phi$ form to a good approximation. For this reason, we can obtain a good overview of the behaviour by studying $\kappa$ for the $\phi$ values where $\kappa$ has extrema, i.e., $\phi = 0$ and $\phi = \pi$, see Fig. 9(a). The linear behavior at low $\Gamma$ is followed by quadratic corrections that are $\phi$-dependent. The splitting increases up to $\Gamma/U \approx 0.2$, in the regime where the impurity spin screening becomes sizeable in the doublet ground state. The splitting then starts to decrease and the $\kappa$ curves cross at $\Gamma/U \approx 0.5$, which is deep in the Kondo screened regime at $\phi = 0$, where the doublet states form the excited state multiplet while the ground state is actually a spin singlet state [39–46]. For very large values of $\Gamma/U \approx 1$ both curves saturate to 1. The emergence of $\phi$-dependence can be better observed in Fig. 9(b) where we plot the $\kappa/\Gamma$ ratio. The non-linearity of $\kappa(\Gamma)$ due to fourth-order hopping processes is clearly visible as the departure from a constant, which is different for $\phi = 0$ and $\phi = \pi$. The magnitude of the $\phi$-dependent part can be quantified through the relative difference

$$\Delta_\kappa = \frac{\kappa_\pi - \kappa_0}{(\kappa_\pi + \kappa_0)/2}, \tag{53}$$

shown in Fig. 9(c). For a very different perspective, we also plot the results as a function of the $E_{DS}(\Gamma)/\Delta$ ratio, where $E_{DS} = E_D - E_S$ is the energy difference between the lowest-lying

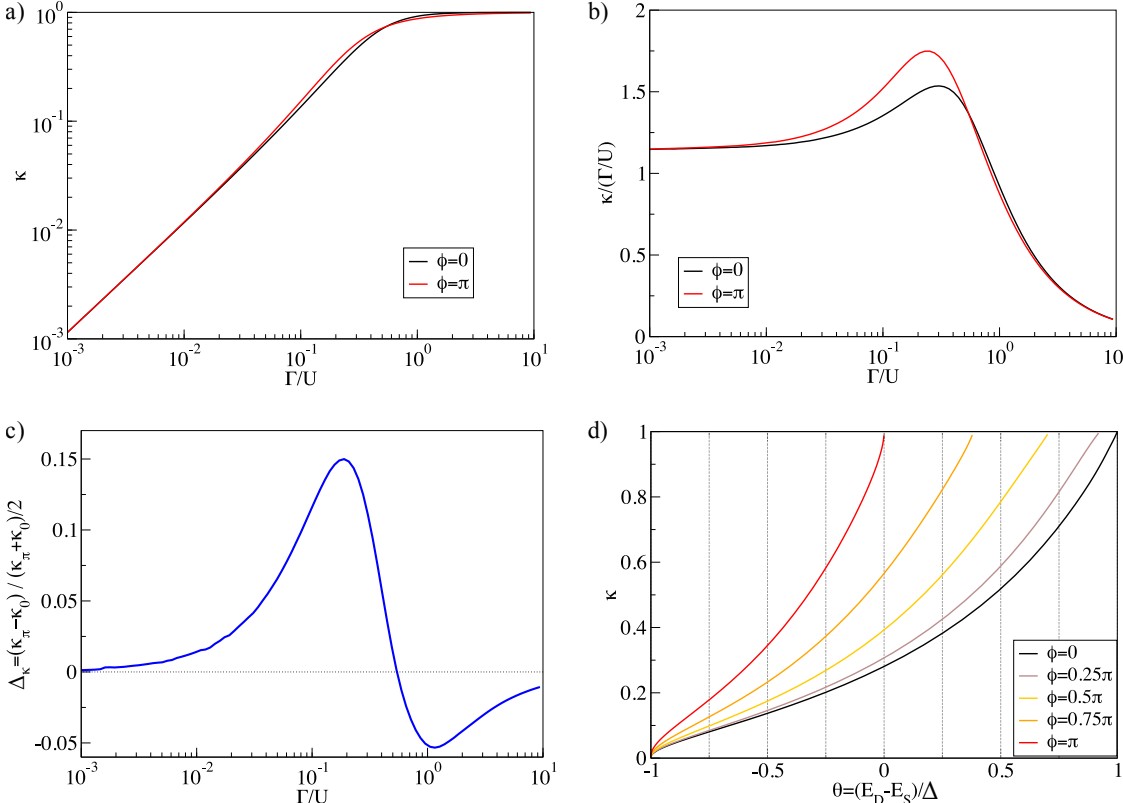

Figure 9: Comparison of the $\Gamma$-dependence of Zeeman renormalisation factor $\kappa$ at $\phi = 0$ and at $\phi = \pi$. **a)** Overview: $\kappa$ vs. $\Gamma$ on log-log scale. **b)** Departures from linearity: $\kappa/\Gamma$ vs. $\Gamma$ on log-linear scale. **c)** Normalized difference of $\kappa$ at $\phi = 0$ and $\phi = \pi$. **d)** $\kappa$ plotted as a function of $\theta = (E_D - E_S)/\Delta$, the ratio of the binding energy (defined as the energy difference between the lowest-lying spin-singlet state and the lowest-lying spin-doublet state) over the SC gap, for a range of $\phi$. The parameters are $U/\Delta = 10$ and $D = 10^2\Delta$.

spin-singlet state and the lowest-lying spin-doublet state, which with increasing $\Gamma$ evolves from $-\Delta$ to $\Delta$ for $\phi = 0$ (this is the well-know behaviour of the subgap states in the SIAM with a SC bath [23]), from $-\Delta$ to $0$ for $\phi = \pi$ (this corresponds to the existence of the "doublet chimney" in the $\pi$-junctions [46, 69, 70]), and from $-\Delta$ to a $\phi$-dependent upper limit for general $\phi$. These line-shapes are further discussed in Sec. 6.6 in the context of universal behaviour (or lack thereof).

Recalling that $\kappa$ is a measure of the degree of spin compensation [14], the results in Fig. 9 reveal that in most of the experimentally relevant range of $\Gamma$ (weak and moderately strong hybridisation) the doublet is more strongly Kondo screened for $\phi = \pi$ compared to $\phi = 0$, i.e., $\kappa(\pi) > \kappa(0)$. This can be intuitively understood as follows. The exchange interaction between the impurity spin and the two SCs by itself does not depend on $\phi$. However, the hybridisation matrix in the Nambu space has an out-of-diagonal component that is proportional to $\Gamma \cos(\phi/2)$. This implies that the proximity effect is strongest at $\phi = 0$, where it leads to a stronger admixture of states where the QD is empty or doubly occupied, at the expense of the singly-occupied configurations that carry the spin degree of freedom. The doublet state thus experiences proportionally weaker Kondo exchange coupling at $\phi = 0$ as compared to $\phi = \pi$. This reduction is proportional to $\Gamma$, while the Kondo coupling $J$ is itself proportional to $\Gamma$, therefore the overall effect is proportional to $\Gamma^2$, as expected.

The regime of large $\Gamma$ can be intuitively understood from the infinite-$\Gamma$ limit. The unper-

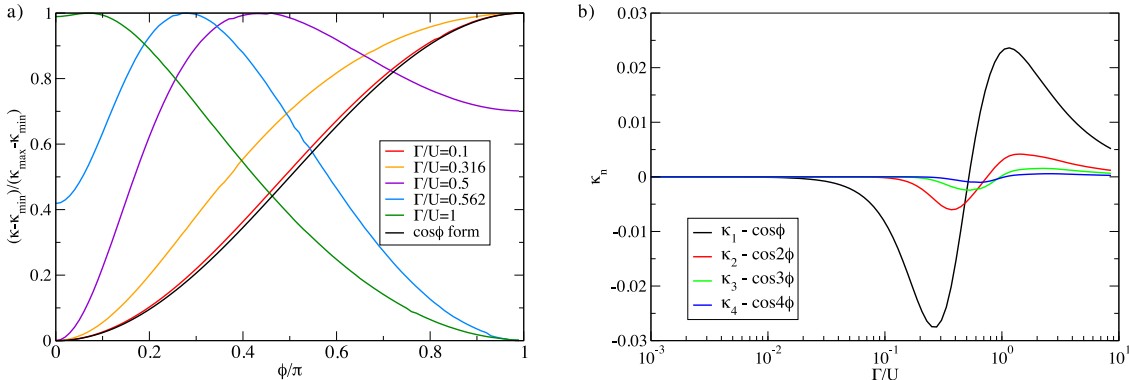

Figure 10: **a**) Phase-dependence of $\kappa$ from weak- to strong-hybridisation regimes. The amplitudes are normalized to emphasize the changing line shapes. **b**) Fourier series coefficients of $\kappa(\phi)$. Parameters are $U/\Delta = 10$, $D = 10^2\Delta$.

turbed Hamiltonian in this case is the normal-state metal with a non-interacting impurity level, while the effects of the Coulomb repulsion on the QD site and of the pairing in the SC leads can be calculated by expanding in $1/\Gamma$. The $\phi$ dependence can be moved from the pairing to the hopping terms using a gauge transformation $c_{\beta,n\sigma} \to e^{-i\phi_\beta/2} c_{\beta,n\sigma}$. In the ground state of the unperturbed Hamiltonian, the impurity is completely absorbed in the continuum and $\kappa = 1$. The expansion in $1/\Gamma$ reveals that at $\phi = 0$ the value of $1 - \kappa$ grows as $1/\Gamma^2$, while for all non-zero $\phi$ the leading correction is linear in $1/\Gamma$. The difference stems from a cancellation of contributions that occurs only for $\phi = 0$, and is hence purely an interference effect. It follows that $\kappa(\pi) < \kappa(0)$.

The change of sign of $\kappa(\pi) - \kappa(0)$ as a function of $\Gamma$ is universal, it happens at any value of $U/\Delta$, away from the particle-hole symmetric point, and also for left-right asymmetric Josephson junctions. This change can be thought to mark the cross-over from the weak-coupling to the strong-coupling regimes that have distinct asymptotic behaviours; for $U \gg \Delta$, the sign change indeed occurs close to $\pi\Gamma/U = 1$ where the perturbation theory breaks down [71–73].

## 6.5 Deviations from pure harmonic form

The plots of $\kappa$ as a function of $\phi$ are shown in Fig. 10(a) for a range of $\Gamma$. At low $\Gamma$, the $\phi$-dependence of $\kappa$ is dominated by the lowest-order (quartic in hopping) terms. Indeed, for weak to moderate $\Gamma$, the curves are close to a perfect $\cos\phi$ function, with only a small deviation at $\Gamma/U$ as large as 0.1. For larger $\Gamma$, including in part of the experimentally relevant regime, the contributions from higher order processes will lead to sizable deviations from the pure harmonic form for $\kappa$. Higher harmonics arise from processes involving the transfer of multiple Cooper pairs. At $\Gamma/U = 0.316$ we observe a 20% admixture of $\cos(2\phi)$ contributions transferring two Cooper pairs, which correspond to processes that are eighth order in electron hopping (the lowest order process where two Cooper pairs hop from one superconducting contact to another). The line-shape evolves rapidly in this range of $\Gamma/U$ and shows very strong deviations from the simple harmonic form. For very large $\Gamma$ the harmonic form is eventually largely restored but with the opposite amplitude of the $\cos\phi$ term. This behaviour is interesting in light of the recent observation that higher harmonics are required for breaking the symmetry between the branches of subgap states, leading to a supercurrent diode effect instead of simple anomalous phase shift in the presence of spin-orbit coupling [74].

We study the evolution of the $\phi$-dependence of $\kappa$ more quantitatively by expanding it in Fourier series up to fourth order:

$$\kappa(\phi) = \bar{\kappa} + \sum_{n=1}^{4} \kappa_n \cos(n\phi). \tag{54}$$

The coefficients $\kappa_n$ are obtained from numerical calculations for $\phi = 0, \pi/4, \pi/2, 3\pi/4, \pi$:

$$
\begin{aligned}
\kappa_1 &= \frac{\kappa_0 + \sqrt{2}\kappa_{\pi/4} - \sqrt{2}\kappa_{3\pi/4} - \kappa_\pi}{4}, \quad \kappa_2 = \frac{\kappa_0 - 2\kappa_{\pi/2} + \kappa_\pi}{4}, \\
\kappa_3 &= \frac{\kappa_0 - \sqrt{2}\kappa_{\pi/4} + \sqrt{2}\kappa_{3\pi/4} - \kappa_\pi}{4}, \quad \kappa_4 = \frac{\kappa_0 - 2\kappa_{\pi/4} + 2\kappa_{\pi/2} - 2\kappa_{3\pi/4} + \kappa_\pi}{8}.
\end{aligned} \tag{55}
$$

We show them in Fig. 10(b) as functions of $\Gamma/U$. The higher harmonics become sizable in the same parameter range where the fundamental changes sign (which happens at $\Gamma/U \approx 0.5$), explaining the complex line-shapes observed in Fig. 10(a). Interestingly, all harmonics undergo a sign change as a function of $\Gamma$ in roughly the same parameter range. This is again a consequence of the crossover from the weak-coupling to the strong-coupling regime.

## 6.6 Departure from universality for $U \sim \Delta$

As opposed to the situation in the Kondo model with no charge degrees of freedom on the impurity site, in the SIAM there is an additional parameter (the ratio $U/\Delta$) that controls the dominant type of charge fluctuations in the impurity problem. In the particle-hole symmetric case, for $U/2 < \Delta$ the lowest-energy charge excitations are local on-site impurity valence changes to zero and double occupancy, while for $U/2 > \Delta$ the lowest-energy parity-changing excitations are Bogoliubov quasiparticles in the SC. For this reason, $\kappa$ is a universal function of $T_K/\Delta$ only in the $\Delta \ll U \ll D$ regime. We explore the deviation from the universality in Fig. 11 where we plot $\kappa$ as a function of $\theta = (E_D - E_S)/\Delta$. With increasing $U/\Delta$, the $\kappa(\theta)$ curves indeed approach the universal curve. The deviation becomes strong for moderate values of $U/\Delta$ approaching 2, with a qualitative change occurring for $U/\Delta = 2$: across this transition point the derivative $d\kappa/d\theta$ at $\theta = -1$, $\kappa = 0$ changes discontinuously from infinite to zero. For smaller $U$, the curves start from the $\theta = -U/(2\Delta)$, $\kappa = 0$ point. The non-analytic behaviour at $U/2 = \Delta$ is a signature of the transition from the regime of proximitized (ABS) subgap states to the genuine Yu-Shiba-Rusinov regime [54].

## 6.7 Asymmetric hybridisation

If $\Gamma_L$ and $\Gamma_R$ are taken to be different, but keeping their sum constant, $\Gamma = \Gamma_L + \Gamma_R$, the results for $\phi = 0$ are unchanged, while the $\phi$-dependence is weakened (results not shown). If we introduce an asymmetry factor $a = \frac{\Gamma_L}{\Gamma_R}$, the amplitude of the $\phi$-dependent part is reduced by a factor that can be established from the mapping presented in Ref. [75]. The results for the asymmetric case can be obtained from those for the symmetric situation with an effective phase shift parameter, so that

$$\kappa(\phi) = \kappa^S \left( 2 \arccos \sqrt{1 - \frac{4a}{(a+1)^2} \sin^2 \frac{\phi}{2}} \right). \tag{56}$$

Here $\kappa^S$ is the impurity Knight shift in the symmetric junction with the same total hybridisation $\Gamma$. In the regime where the phase dependence is harmonic to a good approximation, one can derive from this expression the reduction factor knowing only the numerical results at $\phi = 0$ and $\phi = \pi$; in general, especially close to the strongly anharmonic regime at $\Gamma/U \sim 0.5$, one needs the full $\phi$-dependence.

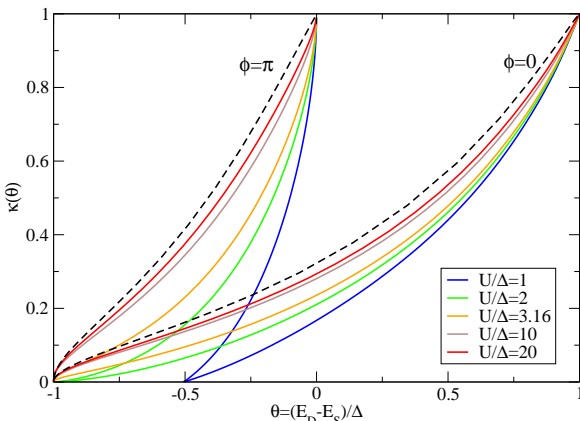

Figure 11: Zeeman renormalisation factor $\kappa$ as a function of the impurity binding energy $E_D - E_S$ normalized by the gap, $\theta = (E_D - E_S)/\Delta$, for a range of $U/\Delta$ showing deviations from the universal behaviour found in the $\Delta \ll U \ll D$ limit (dashed black line). The universal curve has been computed for $U/\Delta = 1000$ and $D = 10^6 \Delta$; in all other cases $D = 10^2 \Delta$.

## 7 Discussion of theory results

### 7.1 Relation between $\kappa(\phi)$ and Josephson energy

We recall that the potential energy of the quantum dot Josephson junction (ignoring SOC for simplicity) is [see Eq. (1)]

$$
\begin{aligned}
U(\phi) &= E_0 \cos\phi \pm \frac{1}{2}[1 - \kappa(\phi)]E_Z \\
&= E_0 \cos\phi \pm \frac{1}{2}\left[1 - \bar{\kappa} + \frac{\Delta_\kappa}{2}\cos\phi\right]E_Z \\
&= \left(E_0 \pm \frac{\Delta_\kappa}{4}E_Z\right)\cos\phi + \dots
\end{aligned}
\tag{57}
$$

This implies that the phase-dependent part of the Zeeman renormalization factor may equally be interpreted as a field-dependent correction to the Josephson energy of the junction. One may hence also write

$$
\Delta_\kappa = -4\frac{\partial E_J(\phi = 0)}{\partial E_Z},
\tag{58}
$$

where the lower level of the spin-doublet multiplet must be taken. This definition is valid in the perturbative (low-$\Gamma$) regime.

### 7.2 SIAM vs. Kondo model

We have shown that the impurity Knight shift factor $\kappa$ depends linearly on the hybridisation strength $\Gamma$ for the single-impurity Anderson model in the weak hybridisation limit. This is in seeming disagreement with the results for the Kondo model [14], where $\kappa \propto J^2$ was found for small $J$. Both results are actually correct, as can be ascertained with explicit NRG calculations. The difference demonstrates that some care is needed in applying the Kondo model

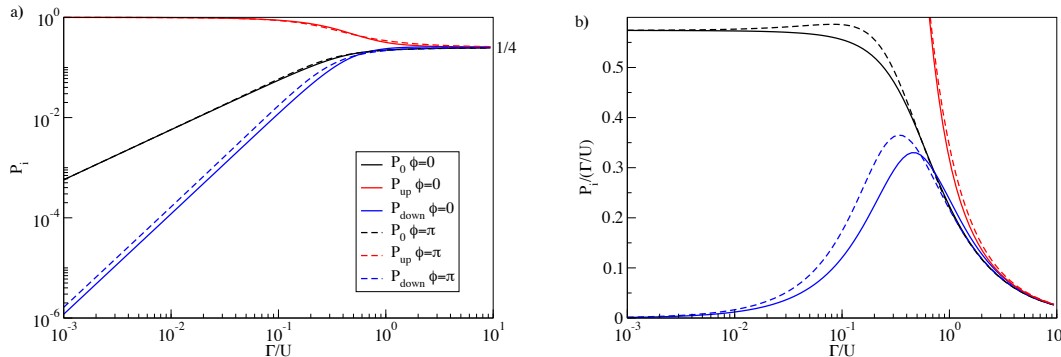

Figure 12: Impurity state (diagonal matrix elements of the impurity density matrix, $P_i$, for the doublet state) as a function of $\Gamma$. **a)** $P_i$ vs $\Gamma$. **b)** $P_i/\Gamma$ vs. $\Gamma$. Due to the particle-hole symmetry, $P_0 = P_2$. The parameters are $U/\Delta = 10$ and $D = 10^2\Delta$.

as an effective model for the SIAM. When one aims for an accurate description of actual experimental setups, the starting point should always be the SIAM with a SC bath, which is a realistic microscopic effective model that adequately describes the low-energy physics of many devices [25,43,48,76,76–78]. The Kondo model may be used as an effective model following a small modification of the Hamiltonian, as we discuss next.

### 7.3 Charge and spin fluctuation mechanisms

To get a more detailed insight into the mechanisms that contribute to the reduction of the Zeeman splitting in SIAM, we study the diagonal matrix elements of the impurity density matrix, calculated for the spin-up ($S_z = +1/2$) doublet ground state; see Fig. 12. These uncover the nature of the wavefunction contributions that lower the state's energy through fluctuations. At $\Gamma = 0$, the completely decoupled impurity spin gives $P_\uparrow = 1$. With increasing $\Gamma$, the spin fluctuations increase the admixture of the doublet excitation with two quasiparticles, where the impurity forms a singlet with one quasiparticle, while the second quasiparticle remains free [70]. The singlet component of the wavefunctions has $P_\uparrow = P_\downarrow$. The contribution of spin fluctuations is thus expressed by the presence of the $P_\downarrow$ contribution. Indeed as shown in Fig. 12(a) in the large $\Gamma$ limit we find $P_\uparrow = P_\downarrow$ and a completely screened impurity spin ($\kappa = 1$). The charge fluctuations are different in nature, and involve states where the impurity is empty or doubly occupied. They are quantified by $P_0$ (which is equal to $P_2$ in case of the particle-hole symmetry).

The most important observation is that the charge fluctuations reduce the local moment (quantified by $S^z_{\text{imp}} = (P_\uparrow - P_\downarrow)/2$) by an amount that is linear in $\Gamma$, while the renormalization due to spin fluctuations is quadratic in $\Gamma$, as inferred from the slopes in Fig. 12(a). In fact, even in the YSR regime with $U = 10\Delta$ the charge fluctuations constitute the dominant contribution to $\kappa$ in a significant part of the parameter range of $\Gamma$. In light of this, for a more realistic description of the effect of the magnetic field in scope of an effective Kondo model for a QD, one should take into account that the spin degree of freedom $\sigma$ in the Kondo model is, in fact, an effective spin variable that labels the two states forming the spin doublet. It is not the same as the physical spin operator $\hat{S}_z = \frac{1}{2}(\hat{n}_\uparrow - \hat{n}_\downarrow)$ in the SIAM which couples with the magnetic field. To relate $\sigma_z$ and $\hat{S}_z$, one should transform the operator $\hat{S}_z$ with the same unitary transformation that is applied to the SIAM Hamiltonian in the Schrieffer-Wolff (SW) transformation [13]. A quick calculation shows that the SW transformation for a normal-state

system maps the impurity spin operator as

$$\hat{S}_z \to e^S \hat{S}_z e^{-S} = \hat{S}_z \left(1 - \sum_\beta \rho J_\beta \frac{D}{U}\right) + \dots, \tag{59}$$

where $S$ is the generator of the SW transformation. We observe that this is precisely the $g$-factor renormalization expected in the $U \gg D$ limit (the limit assumed in the SW transformation [13, 21]), see Eq. (38). This confirms that the leading $\kappa \propto J$ dependence stems from the charge fluctuations in the SIAM. We also note that the $g$-factor renormalization to second order in PT is consistent with the linear order in hopping in the generator of SW transformation (which leads to an effective Kondo exchange coupling which is quadratic in hopping, $J \propto V^2$): both consider the effects of electron excursions from the impurity to the bath to lowest order in hopping events.

Based on these considerations, a Kondo model with a suitable multiplicative correction factor to the Zeeman term is an adequate effective model for studying the spin response of a magnetic impurity. The required correction factor is $1 - \kappa^{(2)} = 1 - \Gamma/(\pi\Delta)i^{(2)}(U/\Delta, D/\Delta)$. Usually $D \gg \Delta$, thus one may use the expression from Eq. (31). The effective Kondo Hamiltonian is hence

$$H_{\text{Kondo}} = H_{\text{band}} + J\mathbf{S} \cdot \mathbf{s}(\mathbf{r}_{\text{imp}}) + g\mu_B \left[1 - \frac{\Gamma}{\pi\Delta} i^{(2)}\left(\frac{U}{\Delta}, \infty\right)\right] BS_z. \tag{60}$$

Here $J = \frac{8\Gamma}{\pi U\rho}$, $\mathbf{S} = \frac{1}{2}\boldsymbol{\sigma}$ is the impurity spin, and $\mathbf{s}(\mathbf{r}_{\text{imp}})$ is the spin density of the conduction electrons at the position of the impurity. If $U < D$, as is usually the case, the bandwidth in $H_{\text{band}}$ should be reduced to an effective bandwidth [21], e.g. $D_{\text{eff}} = 0.192U$ for $U \ll D$.

## 7.4 Ising vs. spin-flip terms

For a Kondo model with an XXZ exchange anisotropy in the limit of pure Ising (longitudinal) exchange coupling, $J_z S_z s_z$, there is no renormalization at all, $\kappa = 0$. It is only the transverse (fluctuating, spin-flip) part $J_\perp(S_x s_x + S_y s_y)$ that leads to the impurity Knight shift. In other words, the impurity Knight shift discussed so far in this work is a dynamical renormalization process, rather than a shift that would follow the static polarization of the electron cloud around the impurity.

## 7.5 Field effects in bulk

Up to this point we have assumed that the magnetic field in the bulk is fully screened by the surface currents within the penetration depth of the superconductor. If the field penetrates the superconductor (e.g. in small SC grains or if the field is applied in plane to a thin SC layer), the quasiparticles will spin polarize [49, 79–81]. Assuming that the magnetic field has no other effect than the Zeeman splitting of the quasiparticle levels, the perturbative calculations from Sec. 5 still apply, but one needs to use spin-dependent quasiparticle energies

$$\xi_{n,\sigma} = \sqrt{\epsilon_n^2 + \Delta^2} + \sigma \frac{1}{2} g_S \mu_B B_S. \tag{61}$$

Here $g_S$ is the atomic Landé $g$-factor of the SC material, $B_S$ the field in the SC baths, and in the last term $\sigma = 1$ for spin up and $\sigma = -1$ for spin down. Assuming weak spin-orbit coupling, all processes conserve $S_z$, thus a transfer of a particle from the impurity to the bath costs $\pm(gB - g_S B_S)\mu_B$ in energy. This can be expressed in several alternative forms:

$$(gB - g_S B_S)\mu_B = \left(1 - \frac{g_S B_S}{gB}\right) g\mu_B B = (1-\tau) g\mu_B B. \tag{62}$$

This implies that all the results derived in this work for $B_S \equiv 0$ remain valid, if $g$ is replaced by $(1-\tau)g$. In particular, the renormalized $g$-factor becomes

$$g_{\text{eff}} = g(1-\tau)(1-\kappa). \tag{63}$$

Thus $\kappa$ itself is unaffected. This is because $\kappa$ has the significance of spin compensation by itinerant electrons, which is by definition a property of the state of the system in the absence of any applied magnetic field, and hence does not depend on the bare Landé $g$-factors of various constituent materials. The measured Knight shift of course does depend on the correction factor $\tau = g_S B_S / gB$. If $B_S \approx B$, as in ultrasmall superconducting grains, $\tau \approx g_S/g$. The relative values of $g_S$ and $g$, and even the signs, vary greatly between devices and even from level to level due to mesoscopic fluctuations, thus it is difficult to make any general statements. In particular, $\tau$ need not be small and there are known cases of QD-SI devices with $|g_S| > |g|$ [48].

## 8  Significance for applications

Before concluding we estimate the magnitude of the $\phi$-dependent effect in realistic situations. We take $B = 100\,\text{mT}$, which is compatible with many superconducting devices, and $g = 15$, a typical value for devices made of III-V semiconductors, which gives $E_Z \approx 90\,\mu\text{eV}$. In Fig. 13 we plot the amplitudes of the $\phi$-dependent part of the impurity Knight shift for several values of $U/\Delta$. Taking the case of $U/\Delta = 3.16$ at $\Gamma/U = 0.2$ (red point in the figure), a fairly typical value at which point the doublet is the ground state of the system (the singlet is at $0.5\Delta$), we find $\kappa_\pi - \kappa_0 \approx 0.026$, that corresponds to $2.4\,\mu\text{eV}$ in energy units or a $580\,\text{MHz}$ frequency shift. This is an experimentally accessible scale using microwave techniques [46, 64]. For devices with higher values of $U/\Delta$, i.e. deeper in the Yu-Shiba-Rusinov regime, even larger shifts can be achieved with the system still in the doublet ground state ($\Gamma/U$ up to $\approx 0.2$, blue points in the figure). The maximal values of $\kappa_\pi - \kappa_0$ seem to be capped to $\approx 0.06$, which corresponds to frequency shifts well in the GHz range. For low values of $U/\Delta$ (in the proximitized state regime) the shifts are smaller because of the larger charge fluctuation and, at the same time, the doublet state is expected to be less long-lived because of the smaller energy differences. Thus for applications aiming to explore the $\phi$-dependent impurity Knight shift, the most appropriate systems are those with a well-defined local moment at large $U/\Delta$.

## 9  Conclusion

We have systematically explored the impurity Knight shift in the single-impurity Anderson model for a QD Josephson junction in all parameter regimes, from weak to strong electron-electron interaction. The leading term in the Zeeman renormalization factor $\kappa$, due to charge fluctuations, is linear in the total hybridisation strength $\Gamma$: each SC lead contributes additively. The subleading term, due to spin fluctuations, contains contributions proportional to $\cos\phi$, where $\phi$ is the gauge-invariant phase difference between the SC contacts, due to Cooper pair transfer processes. This implies a coupling between the operators $\hat{S}_z$ and $\hat{\phi}$, i.e., between the spin and the Josephson current (or the transmon degrees of freedom in the context of Andreev spin qubits and gatemon circuits). The exciting feature here is that the coupling constant is directly proportional to the external magnetic field. This makes the impurity Knight shift useful for control in superconducting spin qubits [82–85]. In particular, the magnetic field enables electric manipulation of spin via electric dipole spin resonance (EDSR) [67, 86–90], and this is the case even in the absence of the spin-orbit coupling because $\kappa$ depends on gate-tunable parameters. We have presented evidence that QD Josephson junctions [35, 46] indeed have

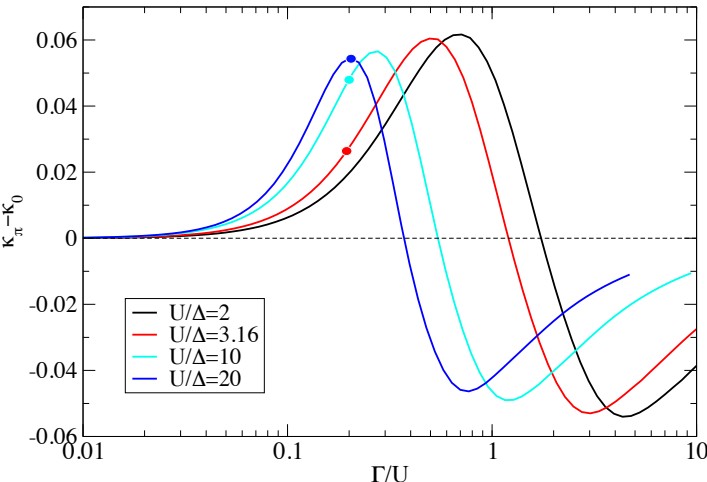

Figure 13: Difference between $\kappa$ values for $\phi = \pi$ and $\phi = 0$ (i.e., the amplitude of the phase-dependent part of the impurity Knight shift) for several $U/\Delta$ ratios. The spin-doublet regime extends up to $\Gamma/U \approx 0.2$. Here $D = 10^2\Delta$.

phase-dependent $g$-factors with a $\cos\phi$ contribution arising from the Cooper pair transfer. The range of materials [91] and device designs [33, 92, 93] where the $\phi$-dependence of the Zeeman splitting could be relevant is wide.

# Acknowledgments

**Funding information**    L. P. and R. Ž. acknowledge the support of the Slovenian Research Agency (ARRS) under P1-0416 and J1-3008. A. B., M. P.-V.: This research is co-funded by the allowance for Top consortia for Knowledge and Innovation (TKI's) from the Dutch Ministry of Economic Affairs, research project *Scalable circuits of Majorana qubits with topological protection (i39, SCMQ) with project number 14SCMQ02, from the Dutch Research Council (NWO), and the Microsoft Quantum initiative.*

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
