# Peer review of "Impurity Knight shift in quantum dot Josephson junctions"

_SciPost Physics, doi:SciPost Phys. 15, 070 (2023)_

## Round 1 · Referee Report · Volker Meden (Referee 1) · 2022-12-27

Strengths

1-timely manuscript which makes contact to a very recent paper on the same question for the Kondo model (here the single-impurity Anderson model) as well as to experimental data
2-comprehensive presentation including perturbation theory (in the impurity-reservoir hybridization) and (accurate) numerical for results beyond the perturbative regime (which includes the parameter regime relevant to most experiments)
3-overall well written (for suggestions; see below)

Weaknesses

The only weaknesses I found are of detailed nature. I will list them below.

Report

In the manuscript the change of the impurity g-factor due to the coupling of the impurity to two (BCS) superconductors is analyzed theoretically in due detail. This is a question of current theoretical and experimental interest. The manuscript is directly related to recent theoretical and experimental publications. It clarifies the role of charge fluctuations (Kondo model versus single-impurity Anderson model) and discusses the dependence of the g-factor change (Knight shift) on the phase difference between the two supercondutors. I am confident that the present work is of current interest, clarifies a pressing reserach question, and will lead to follow-up publications. The presentation is overall very clear. The results are discussed in a transparent form. I can thus recommend publication. However, I stumbled accross a few detailed issues the authors might want to revise to even enhance the clarity of the presentation. I'll gives these below.

Requested changes

This is a list of suggestions. It should be obvious which ones are minor (e.g. typos or so) and which ones are more important. 1-page 4, first paragraph of Sect. IV: "...using the projector operator approach AND a symbolic algebra system..." 2-Notation Eqs. (15), (17), (18): I suggest to use the bracket-notation also for the states appearing on the left hand sides of these equations. I was inintially confused by this switch of notation. This appears to be unnecessary. 3-page 6, right before Eq. (25): Speaking here and following Eq. (30) of U<<Delta is confusing as what is given is in fact the U=0 result. In particular, no corrections in oders of U/Delta are given. 4-Fig. 3: It woulld look nicer if there would be a small space between the "I" and the upper index "(4a/b)". 5-page 7, right below Fig. 3: What is meant by "...the same rule as in the previous section..."? The argument employing the intermediate states? 6-Fig. 4: I suspect that the gray lines again indicate U=Delta and U=D? Please clarify. I am confused by the caption. I can only see red dasehd lines showing the asymptotics at small U!? 7-page 9, last line: The sentence "The leading Γ^2 correction better captures the deviation from linearity..." is confusing. The Gamma^2 term IS the leading correction to linearity and thus must be included to capture any deviation from linearity. Furthermore, "...underestimate the renormalization in the intermediate Γ regime." What exactly is the "intermediate Gamma regime"? 8-page 10, second paragraph: I am puzlled by the discussion of "...clearly visible saturation effects...". I do not recognize any saturation in the two plots of Fig. 10 while the linearity in the corresponding variables as described in the text is obviuos. 9-Fig. 8 and last sentence of Sect. V.B: Is there a typo in the x-axis label? As I understood it D is always set to 1. I suspect that the x-axis label is U/Delta? In that case the last sentence of Sect. V.B appears to be meaningful as well (otherwise not). 10-Sect. V.D, bottom of page 11.: As the doublet-singlet quantum phase transition is not discussed in the present manuscript as reference might be useful. 11-Sect. V.D: Inclusing a single subsubsection "1...." is confusing. Just drop the "1....". 12-page 13: How do you know that the cos(2 phi) term is fourth order in Gamma? Please elaborate. 13 - Sect. V.F: I do not understand why the attempt to capture the Theta dependence of kappa in a phenomenological way (Eqs. (439-(46)) is meaningful or useful. Please elaborate.

---

## Round 1 · Referee Report · Anonymous (Referee 2) · 2023-2-8

Strengths

1-Significantly extends a new direction in the research field of superconducting quantum dots, namely the Knight shift and spin response to microwave drives
2-Systematic wide scope study of various aspects of the problem including massive NRG calculations, perturbation theory as well as relation to the experiments

Weaknesses

1-Assembly of many interesting and relevant aspects of the studied model, which, however, are not too well bound into a single coherent story
2-Presentation of results especially in the perturbation theory part of the manuscript
3-Several confusing inconsistencies throughout the text

Report

The paper studies the Knight shift in a quantum dot coupled to two phase-biased superconducting leads and thus forming a nanoscopic Josephson junction. This topic is fresh, motivated by a recent experiment and the paper thus satisfies one of the necessary criteria of acceptance "Open a new pathway in an existing or a new research direction, with clear potential for multipronged follow-up work;" and deserves publication in SciPost Physics.

On the other hand, its current form is not quite suitable for direct publication and should be further improved as I will detail below. Apart from a number of minor suggestions I see two global shortcomings.
First, I was really disappointed when finally coming to Sec. VII about the experimental evidence. The data were analyzed to show the existence of a term linear in the magnetic field and proportional to the cos of the phase difference in line with the Knight shift assumption but not even an attempt to apply the theory developed on the previous 17 pages had been made. In this respect the experimental part should be taken solely as a motivation of the otherwise purely theoretical study, i.e., this is not in my view a joint theory-experiment paper, which was my impression when reading the abstract.
Second, it's not very clear to me why the authors spent so much effort on the perturbation theory since its connection to the nonperturbative NRG results is not too elaborated in the text and, moreover, it's claimed anyway that PT is not of much relevance for the experiments which are typically in the nonperturbative regime.
What I miss is a more coherent picture unifying all essential ingredients, i.e., perturbation theory, NRG, and experimental findings.

I am not sure if and how this could be improved but it would definitely help the paper as well as the readers.

Requested changes

Here I list various questions, comments, and suggestions for authors' consideration:
1-\beta-index missing in Eqs. (8), (17), and (18). Matrix elements m_A/B,n\sigma in Eqs. (17) and (18) not defined nor mentioned anywhere else.

2-Horizontal axes in Figs. 2, 3, and 4 are not in dimensionless units (in other graphs they are).
Very misleading/confusing caption in Fig. 2 ("U relative to the gap \Delta AND (?!) the bandwidth D") .
Problems with caption of Fig. 3: (top) - wrong?, grey lines nearly invisible in my printout, missing = in the last line.
Problems with caption of Fig. 4: I don't see any RED dashed lines, just some kind of orange rather close to the orange full line, moreover NO dashed line for large U mentioned in the caption. Reference to a mysterious logarithmic behavior, see also below.
Misprint in "bandwidth" between Eqs. (25) and (26).

3-General reservations to handling and presenting perturbative results.
Perturbation theory, unlike NRG, provides (semi)analytical results and, thus, the depiction and presentation conventions are somewhat different from purely numerical, e.g., NRG results. It's a good practice to present results as functions of a minimal number of dimensionless parameters, i.e., the potentional dimesionful prefactor is extracted and only dimensionless integrals are analyzed. In Eq. (23) this would correspond to writing (dimensionless by its nature) kappa^(2) as Gamma/Delta x dimensionless integral depending on two dimensionless variables U/Delta and D/Delta.
Furthermore, very much unlike NRG, the bandwidth is not the most important and thus the reference parameter. It's far more natural to choose as the energy unit the BCS gap Delta. Then the bandwidth can be easily sent to infinity which is a generic analytical approach also of broad experimental relevance. In fact, I have never seen D being fitted from the experiment (I don't say it's impossible, just not very common I guess), while Gamma, U, and Delta are routinely (attempted to be) extracted. You write yourselves that the narrow-band limit D<<U, where D is important, "is mostly of academic interest" [just above IV.C]. The integral I^(2) in Eq. (23) can be straightforwardly analytically calculated even for finite D and in the limit of D->infty one just gets a single dimensionless function of the dimensionless parameter U/Delta with a clear asymptotic behavior. For finite D, there is the D-dependent truncation for large values of the parameter. It can be intuitively wrapped into one single plot (a hopefully self-explanatory skeleton example attached); making it into a three panel figure (moreover for a fixed D) is in my view just a confusing overkill.
Similarly, just worse, for the fourth order. Here, you don't even present the formulas defining the integrals in Eq. (33). It's then quite hard (to try) to reproduce your results, which directly contradict the SciPost requirement "Provide sufficient details (inside the bulk sections or in appendices) so that arguments and derivations can be reproduced by qualified experts;" of the general acceptance criteria. Again, it would be nice to make the integrals dimensionless by pulling out 1/Delta^2 factor. That would decrease your (ideally small) pertubative correction by eight orders of magnitude from 10^11 to about 1000. Yet, it's still a big number and one might question what the status of the perturbative expansion actually is when the fourth order gives a factor about 1000 times bigger than the second one. Does this order of PT converge at all for infinite D? I can't answer those questions since you don't give any formulas but I would love to get the answers. You also mention some logarithmic correction (whose origin is completely unclear) which miraculously adds up with a 1/U^2 asymptotics of the 4b contribution to a constant plateau [just above IV.C.3]. Furthermore, you study asymptotics of large U>>D (i.e., in the narrow-band limit) which you claimed to be "mostly of academic interest" in just previous subsection. However, I can't see anywhere the far more interesting, generic, and thus relevant D->infty limit.
I believe that these issues should be addressed far more thoroughly to bring solid understanding of the perturbative behavior of the model.
Two more particular questions to this part:
i. Why the curve in Fig. 3f grows for small U while it plunges in Fig. 3e? I roughly understood that compared to the above panels c) and d) you basically multiply the curves by U^2 or U^4, respectively which should make them both fall down for small U, the one in f) even faster. Where am I wrong?
ii. In Fig. 5b you plot the NORMALIZED difference of kappa^(4). Why normalized, why this way? Would this quantity be of any experimental relevance (my answer is "absolutely not"). Experimentally meaningful quantity is just the unnormalized difference which you in fact consider in Fig. 15. Presentation of the normalized difference is in my view redundant if not misleading...

4-In V.A I don't understand the term "saturation", which is in the text related to the small-Gamma asymptotics while in the title it seems to be its opposite. I can't observe "a near perfect overlap of the curves", sorry, I don't understand what's meant by that.
Why the legend in Fig. 7(left) changed on the fly from U/Delta to U/D?
I kind of grasp the explanation of the saturation in Sec. V.B but why the curves for very small Gamma/U don't also saturate for large U?
Missing = signs in caption of Fig. 9.
Don't you mean Gamma cos(phi/2) in V.D.1? Otherwise, I completely miss the justification...

5-Sec. VI:
I suggest to cite PRL 108, 227001 (2012) by Luitz et al. in the context of relevance of SC-SIAM for modelling experiments since this was one of the main conclusions of that paper which precedes all the currently mentioned references.
I don't understand Fig. 13 and the explanations just below it (first two paragraphs). It's perhaps obvious to the NRG people but certainly not to a nonspecialized reader. Was the impurity density matrix in the figure calculated to for a finite applied magnetic field (since the occupations of spin up and down are dramatically different)? What's the value of the field? What happened to the red curve in the right panel? Could you describe what's seen in the figure before you start writing about things which are not there? I like the idea of the connection between the Kondo and SC-Anderson model and the different asymptotics of kappa in the two models. However, I would need a last sentence (here missing) to understand the implications of Eq. (49) for the J^2 behavior. Could you be more eloquent?

6-Finally, in the first line on p. 19 you write that "For devices with higher values of U/Delta...even larger shifts can be achieved...". Doesn't Fig. 15 show the opposite?

Attachment

---

## Round 2 · Referee Report · Anonymous · 2023-4-5

Report

I will basically just comment on the changes made in the updated version.

At the first sight, it seemed that authors made the required changes and the updated manuscript is now fine. However, when plunging more into details especially in the analytical section V on the perturbative approach I have to my disappointment discovered that only cosmetic changes were performed and most of the issues remained. Some of my comments were just completely ignored and I still find the presented results very problematic for several reasons.

First, authors insist on not revealing the integrals for the fourth order contribution and refer to the Mathematica notebook instead. I don't consider Mathematica notebook to be an equivalent replacement of a formula, even if potentially long and complicated but OK. So I went to the repository and downloaded that half-a-gigabyte large zip archive. Unfortunately, my computer somehow hasn't managed to unpack the beast (tried twice) so that I have not been able to see the formulas anyhow yet. My computer is not particularly old or weak so I don't know what to take from that.

Second, the wide band limit is claimed in general inaccessible by the authors but I cannot agree with that. It's likely and would be very useful to show it explicitly that the sum of the two fourth-order contributions has a finite limit when D goes to infinity as it was clearly demonstrated in Eqs. (43) and (44) for the U=0 case. The integration can be performed numerically even for infinite integration limits as long as the integrand decays fast enough. That's exactly the reason why I would need to see the formulas... Instead, the "academic" U>>D limit results are presented and proclaimed in the answer to be "a reasonable approximation for the Delta<U<D regime", although this is not true even in the second order, where it gives qualitatively different results, compare Eq. (35) with (37) .

There are other questions concerning the validity and usefulness of the presented analytical approach and results. For example, particle-hole symmetric case is always considered in Sec. V. Does the limit U->infty then make any physical sense? I believe that U=infty is a useful approximation under suitable conditions, but then one typically considers finite value of epsilon (single-particle level energy). Sending both energies to infty in my view gives just another purely academic answer with no practical implications.

I am not sure, but isn't the phi-dependent part of kappa given by the magnetic field derivative of the critical current through the junction? If so, it would be useful to explicitly mention it. Then another question arises concerning results in Eq. (42). The critical current of a noninteracting resonant level (epsilon=U=0) diverges within the cotunneling approximation (easily seen - the critical current as a function of epsilon is given by a Lorentzian curve; at the resonance one gets 1/Gamma, i.e. the perturbation theory is singular at this point). This points at a possible issue with the order of various limits (U=0, epsilon=0, B=0) and should be handled with big care...

I don't understand the curves in Figs. 4 and 5 for small U. According to Eq. (43) U=0 limit gives for D=10^4 roughly -275 for the a-contribution and 355 for the b-contribution. Their sum (the physically relevant result) is about 80. I am not asking about the logarithm of a negative number in Fig. 4a (I guess the absolute value was taken), but I really don't understand how the summed up contribution (about 80) in Fig. 5a can be above the partial ingredients (around 300) [I am aware of the multiplication by x^2 and logarithmic axes but still, it should be below the two].

BTW, it would be good to explicitly refer to the corresponding analytical expressions for large and small U in the caption of Fig. 5, they are not immediately obvious. What is "r" in Eq. (45)?

I find a bit outrageous the uncorrected statement about the "logarithmic behavior, which is best seen on the log-linear scale in Fig. 5b" (top of p. 10). The authors refer to integrals which were sadly never revealed anywhere so that they are hard to appreciate. Moreover, one should see a straight line in the range (1,10^4) but there are not straight lines (blue and yellow).

I do agree with the authors that "the perturbation theory approach is valuable as it gives physical intuition into the type of processes that produce the corrections. In our case this gives the important insight that the coherent fourth-order pair transfer is the culprit for phi-dependent kappa".

Therefore I strongly suggest:
1. (Physics) Elucidate what is the relation of the phi-dep. kappa and the supercurrent.
2.(Math, Formal) Correct and enhance the analytical perturbative results. In particular, please provide in an accessible way the formulas behind the fourth order and perform their reliable analysis.

---

## Round 2 · Author Response

We would like to thank the editor and both referees for their very constructive feedback. We feel that it helped us greatly improve the manuscript.

Taking the suggestion from Referee #2, we changed the way the theoretical and experimental results are related. The measurements are now presented at the very beginning and serve as the motivation for the calculations that follow. Furthermore, we have significantly improved the presentation and the discussion of results in the perturbation-theory (PT) sections. In the section on fourth-order PT, the focus is now on the physical quantities as opposed to the properties of each contribution which by themselves do not carry much physical information. We only managed to solve the I^(4) integrals analytically in U=0 and U->\infty limits. To obtain the D>>U,Delta limit one would have to analytically solve the full integral and then calculate the limit of large D, which seems unfeasible. Nevertheless, it seems that U=0 results allow to estimate the renormalization factor kappa^4 for finite U << Delta, while the U=\infty result is a reasonable approximation for the Delta < U < D regime.

The perturbation theory approach is valuable as it gives physical intuition into the type of processes that produce the corrections. In our case this gives the important insight that the coherent fourth-order pair transfer is the culprit for phi-dependent kappa. This is thus an important part of the manuscript and needs to be included. On the other hand, the numerical calculations (using NRG) provide accurate numerical results with no parameter restrictions. Comparing the two (like in Fig. 7) quantifies the regime where understanding the problem using simpler low-order processes is valid and shows where higher-order processes become important.

We hope that we have improved the clarity of the manuscript and that it now fulfils all criteria for acceptance.

---

## Round 2 · List of Changes

- Changed the wording in the abstract and introduction to better reflect the fact that the experimental data are used as motivation for theoretical developments.
- Moved the Experimental evidence section to the front.
- Explicitly wrote out the matrix elements in and around Eq. (20), (21).
- Reduced Fig. 3 (previous Fig. 2) to a single panel with dashed lines for limits.
- Rewrote the discussion in Sec. V B. The integral is now expressed with dimensionless parameters (eq. (28)).
- Clarified the matrix elements in Eq. (39). The full expressions of I^(4a) and I^(4b) are very long and we feel writing them explicitly would not contribute to the manuscript. They are however available in a Mathematica Notebook.
- Rewrote the discussion and expressions in Sec. V C.
- Removed panels c-f of Fig. 4. Their only purpose was to confirm the agreement of the analytically obtained limits to the numerical integration.
- Fixed colors in Fig. 5.
- Fig. 6(b) now shows the absolute difference between \kappa at \phi=0 and \phi=\pi.
- Cleared up the confusion regarding the word "saturation" in Sec. VI A and regarding Fig. 8.
- Replaced the reference to the singlet subgap state in Sec. VI D. Now we refer to spin screening in the doublet ground state.
- Removed the subsubsection VI D 1.
- Fixed the cos(phi) -> cos(phi/2) typo.
- Sec. VI E; cleared up the reasoning for the order of higher harmonics.
- Removed the discussion of \kappa(\theta) in Sec. VI F.
- Added a reference to PRL 108, 227001 (2012) in Sec. VII A.
- Added the P_up line to Fig. 14(b).
- Improved the explanation of the physical relevance of different P in Sec. VII B.
- Added the expression for the correction factor necessary for the Kondo model, below eq. (51).
- Added a few words in Sec. VIII to clarify Fig. 15.

Resubmission 2212.07185v3 on 19 May 2023

You are currently on this page

Resubmission 2212.07185v2 on 2 March 2023

---

## Round 3 · Author Response

We would like to again thank the referee for constructive feedback, which has helped us to remove the remaining issues with the manuscript.

Regarding the Zenodo repository:
We have not found anything wrong with the archive, the compatibility issue might indeed be related to the file size or with the use of tgz format, which might be problematic on some computer platforms. For this new version, we provide a new set of files in a more commonly used zip format, one per figure, thereby also fixing the problem of large files, since only some datasets are large. The Mathematica notebooks are, in fact, small files.

Regarding the manuscript:
We thoroughly reworked Section V C on the fourth-order perturbation calculations. The integrals are now explicitly given in terms of quasiparticle energies in a somewhat compact form; we apologise for not providing those in the body of the manuscript in the previous versions. Also we would like to thank the reviewer’s insights which has helped us to correct a sign issue in the definition of the fourth-order correction (relative sign of a vs. b terms). The main results are not affected by this sign change in a significant way since the cos(phi) term had the correct sign. Nevertheless, fixing the sign issue is important because the phi-independent fourth-order correction actually changes sign at U somewhat above 2Delta. In fact, this sign change can be observed in the change of curvature in the NRG results shown in Fig. 6a (new numbering in revised version): the curves for small U curve downwards (when kappa^4 is negative), while they curve upwards at large U (kappa^4 > 0). The fourth-order results we report are now correct and the perturbation matches the low-Gamma numerics slightly better (Fig. 5, new numbering).

We removed the expressions for the U->infty limit at finite bandwidth, as we agree that they perhaps indeed do not contribute all that much to the physical understanding. In their place, there is now a discussion of the integrals in the wide-band limit. The two fourth-order contributions are no longer considered separately, as this way of splitting the processes comes from the mathematical formulation of the perturbation theory and it is thus somewhat arbitrary. Instead, we chose to discuss the phi-dependent and the phi-independent processes separately. We were not able to find closed-form expressions for the integrals, but we present the numerical results in the form of a product of the leading U/Delta-dependence multiplied by a correction function of order 1 (for U/Delta in the usual range). These correction functions are shown in the new Fig. 4. We also remark that to obtain well convergent integrals for numerical evaluation, we symmetrized the integrand with respect to one of the integration variables to cancel out a problematic asymmetric slowly converging term.

Regarding the U->infty limit (such that epsilon/U remains constant): While academic, it is commonly used in the context of mapping between the SIAM and Kondo models (Schrieffer-Wolff transformation) and thus very important in the theory of quantum impurity models. Here epsilon/U is fixed so that the quasiparticle scattering phase shift remains constant when taking the limit. But we agree that this limit is less relevant as regards the analysis of experimental results.

We added a brief discussion (Sec. VII A) on the relation between kappa and Josephson energy, which is itself proportional to the supercurrent.
We agree with the referee that one could obtain kappa by taking the field derivative of the critical supercurrent. However, in our formulation kappa is directly calculated and therefore we avoid any problems arising from taking derivatives of potentially diverging quantities. In particular, kappa does not appear to be singular for epsilon=U=B=0.

We hope that the improvements to the discussion of fourth-order perturbation theory are satisfactory.

---

## Round 3 · List of Changes

• reworked section V C
  • added a sentence on the U->infinity limit
  • added a discussion on the relation between kappa and Josephson energy (Sec. VII A)

---

## Editorial Decision

published